# Na$_v$1.7 as a chondrocyte regulator and therapeutic target for osteoarthritis

Wenyu Fu[1,2], Dmytro Vasylyev[3,4], Yufei Bi[1], Mingshuang Zhang[1], Guodong Sun[1], Asya Khleborodova[1], Guiwu Huang[1,2], Libo Zhao[1,2], Renpeng Zhou[1,2], Yonggang Li[1,2], Shujun Liu[3,4], Xianyi Cai[1], Wenjun He[1], Min Cui[1], Xiangli Zhao[1,2], Aubryanna Hettinghouse[1], Julia Good[1], Ellen Kim[1], Eric Strauss[1], Philipp Leucht[1], Ran Schwarzkopf[1], Edward X. Guo[5], Jonathan Samuels[6], Wenhuo Hu[7,8], Mukundan Attur[6], Stephen G. Waxman[3,4 ✉] & Chuan-ju Liu[1,2,9 ✉]

Osteoarthritis (OA) is the most common joint disease. Currently there are no effective methods that simultaneously prevent joint degeneration and reduce pain[1]. Although limited evidence suggests the existence of voltage-gated sodium channels (VGSCs) in chondrocytes[2], their expression and function in chondrocytes and in OA remain essentially unknown. Here we identify Na$_v$1.7 as an OA-associated VGSC and demonstrate that human OA chondrocytes express functional Na$_v$1.7 channels, with a density of 0.1 to 0.15 channels per μm$^2$ and 350 to 525 channels per cell. Serial genetic ablation of Na$_v$1.7 in multiple mouse models demonstrates that Na$_v$1.7 expressed in dorsal root ganglia neurons is involved in pain, whereas Na$_v$1.7 in chondrocytes regulates OA progression. Pharmacological blockade of Na$_v$1.7 with selective or clinically used pan-Na$_v$ channel blockers significantly ameliorates the progression of structural joint damage, and reduces OA pain behaviour. Mechanistically, Na$_v$1.7 blockers regulate intracellular Ca$^{2+}$ signalling and the chondrocyte secretome, which in turn affects chondrocyte biology and OA progression. Identification of Na$_v$1.7 as a novel chondrocyte-expressed, OA-associated channel uncovers a dual target for the development of disease-modifying and non-opioid pain relief treatment for OA.

Osteoarthritis is a disabling, degenerative disorder distinguished by progressive joint failure[3]. Although it is currently unclear whether the primary cause of OA is cartilage damage, OA always involves cartilage breakdown and loss of the unique extracellular matrix that normally guarantees the compressive resilience essential for joint function[1]. OA chondrocytes undergo complex changes, including anabolic and catabolic alteration. Chondrocytes are central protagonists in this regulatory cascade—as the target of external biomechanical and biochemical stimuli, as well as the source of proteases, cytokines and mediators that regulate the deterioration of articular cartilage[4]. Despite the high prevalence and morbidity of OA, effective disease-modifying treatments are not currently available, and the molecular mechanisms involved in OA remain poorly understood.

Alongside significant loss of articular cartilage, the dominant clinical symptom of OA is pain[5]. Specialized peripheral sensory neurons are abundant in joint tissues, including synovium and subchondral bone[6], and contribute to pain in OA. These neurons express unique repertoires of VGSCs[7]. There are nine distinct VGSCs (Na$_v$1.1–Na$_v$1.9), encoded by genes *SCN1A*–*SCN11A*[8]. Na$_v$1.7, Na$_v$1.8 and Na$_v$1.9 are of particular interest as targets for pain treatment owing to their preferential expression in

peripheral sensory neurons within dorsal root ganglia (DRG), and their roles in action potential initiation and propagation within peripheral pain pathways[7]. Modulation of DRG-expressed Na$_v$1.8 can attenuate OA pain[9]. The critical role of Na$_v$1.7 in pain signalling[10] and genetic validation (severe pain with gain-of-function Na$_v$1.7 mutations[11,12] and insensitivity to pain with loss-of-function Na$_v$1.7 mutations[13,14]) have further supported Na$_v$1.7 as a therapeutic target for pain. Notably, Na$_v$1.7 gain-of-function mutations increase pain sensitivity in some patients with OA[15]. A role of Na$_v$1.7 in inflammatory pain is supported by observations in global Na$_v$1.7 and DRG-specific knockout mice[16,17]. A role of DRG-expressed Na$_v$1.7 in OA pain was supported by reduced OA pain following spinal administration of ProTx II, a Na$_v$1.7-selective antagonist, in the monosodium iodoacetate (MIA)-induced model of OA[18].

Although expression of VGSCs in excitable cells is well-known[8], they have also been observed in cell types that are not considered electrically excitable, including astrocytes, microglia, macrophages and cancer cells[19]. Cartilage is avascular and aneural, but angiogenesis and sensory nerve growth into OA cartilage may contribute to OA pain[20]. In addition, the presence of tetrodotoxin (TTX)-sensitive VGSCs in rabbit chondrocytes has been reported[2]. Nevertheless, although there is

[1]Department of Orthopaedic Surgery, New York University Grossman School of Medicine, New York, NY, USA. [2]Department of Orthopaedics and Rehabilitation, Yale University School of Medicine, New Haven, CT, USA. [3]Department of Neurology, Yale School of Medicine, New Haven, CT, USA. [4]Center for Neuroscience and Regeneration Research, Veterans Affairs Connecticut Healthcare, West Haven, CT, USA. [5]Department of Biomedical Engineering, Columbia University, New York, NY, USA. [6]Division of Rheumatology, Department of Medicine, New York University Grossman School of Medicine, New York, NY, USA. [7]Human Oncology and Pathogenesis Program, Memorial Sloan Kettering Cancer Center, New York, NY, USA. [8]Marie-Josée and Henry R. Kravis Center for Molecular Oncology, Memorial Sloan Kettering Cancer Center, New York, NY, USA. [9]Department of Cell Biology, New York University Grossman School of Medicine, New York, NY, USA. ✉e-mail: stephen.waxman@yale.edu; chuan-ju.liu@yale.edu

evidence that the aberrant activation of VGSCs contributes to OA pain[21], the presence and function of VGSC(s) in chondrocytes and their roles in OA progression and pain remain essentially unknown.

To identify novel, differentially expressed genes in OA, we performed RNA-sequencing (RNA-seq) analysis on normal and arthritic cartilage, and identified $Na_v1.7$ as the only significantly upregulated OA-associated VGSC. Here we demonstrate that distinct from DRG-expressed $Na_v1.7$, which is only involved in OA pain signalling, chondrocyte-expressed $Na_v1.7$ regulates chondrocyte biology and OA progression. We demonstrate that $Na_v1.7$ blockade protects joints from deterioration, and $Na_v1.7$ blockade mediates its chondroprotective effects at least in part by regulating intracellular $Ca^{2+}$ signalling and the chondrocyte secretome.

## $Na_v1.7$ in chondrocytes, elevated in OA

We assessed the expression profile of VGSCs in human chondrocytes using PCR with reverse transcription (RT–PCR) and found that, *SCN2A*, *SCN3A*, *SCN4A*, *SCN8A*, SCN*9A* and *SCN11A* are expressed in chondrocytes (Extended Data Fig. 1a). In line with this data, analysis of the genes that are differentially regulated between OA cartilage (Kellgren–Lawrence (KL) grade 3–4) and non-arthritic cartilage using our previous RNA-seq dataset[22] (GSE168505) with a particular interest in VGSCs, indicated that six VGSCs were expressed in chondrocytes (Extended Data Fig. 1b). However, $Na_v1.7$ mRNA (encoded by *SCN9A*) was the only VGSC transcript that was prominently upregulated (2.69 fold increased, $P < 0.05$) in OA cartilage compared with non-arthritic cartilage (Extended Data Fig. 1b). Among six VGSCs expressed in human chondrocytes, *SCN9A* expression was significantly induced by TNF and IL-1β, pro-inflammatory cytokines associated with OA[4] (Extended Data Fig. 1c). Quantitative PCR with reverse transcription (RT–qPCR) analysis of $Na_v1.7$ expression using mRNAs obtained from independent cartilage specimens (11 non-arthritic, 14 KL grade 1–2 OA and 22 KL grade 3–4 OA) confirmed that the $Na_v1.7$ mRNA expression was significantly increased in both KL grade 1–2 and KL grade 3–4 OA cartilage compared with non-arthritic cartilage (Extended Data Fig. 1d). $Na_v1.7$ protein level was also increased in all grades of radiographic severity (KL 1–4) of human OA compared with non-arthritic controls (Extended Data Fig. 1e). Membrane localization of $Na_v1.7$ in chondrocytes was confirmed following fractionation of human chondrocytes and western blotting by using cytosolic and membrane fractions (Extended Data Fig. 1f). Immunohistochemical staining demonstrated increased expression of $Na_v1.7$ in both human and mouse OA cartilage (Extended Data Fig. 1g,h). Collectively, these data demonstrate that $Na_v1.7$ is expressed in chondrocytes and associates with OA progression.

## OA chondrocyte $Na_v1.7$ electrophysiology

We assessed the presence of $Na_v1.7$ currents in 77 human chondrocytes isolated from three patients with OA (Supplementary Table 1). We first used 1 μM TTX, which blocks all $Na_v$ channels except $Na_v1.5$, $Na_v1.8$ and $Na_v1.9$, to isolate TTX-sensitive (TTX-S) currents which were obtained by subtraction of TTX-resistant (TTX-R) currents from currents in control solution at the respective voltages. Figure 1a shows representative current traces in response to −50 mV, −30 mV, −10 mV and 10 mV test pulses in control solution (Fig. 1a, left), in the presence of 1 μM TTX (Fig. 1a, middle), and the resulting traces of TTX-S current (Fig. 1a, right). Sodium currents were elicited by test voltages (−60 mV to 50 mV in 10 mV increments) applied from −90 mV holding potential. Because the current amplitude was relatively low, we enhanced the signal-to-noise ratio by averaging eight runs for each current–voltage trial. Consistent with a substantial contribution of $Na_v1.7$, TTX-S current began to activate at −40 mV threshold, and exhibited maximal peak amplitude at 0 mV and sodium current reversal potential ($E_r$) = −60.6 ± 2.3 mV ($n = 3$) (Fig. 1b). $G/G_{max}$ of TTX-S current

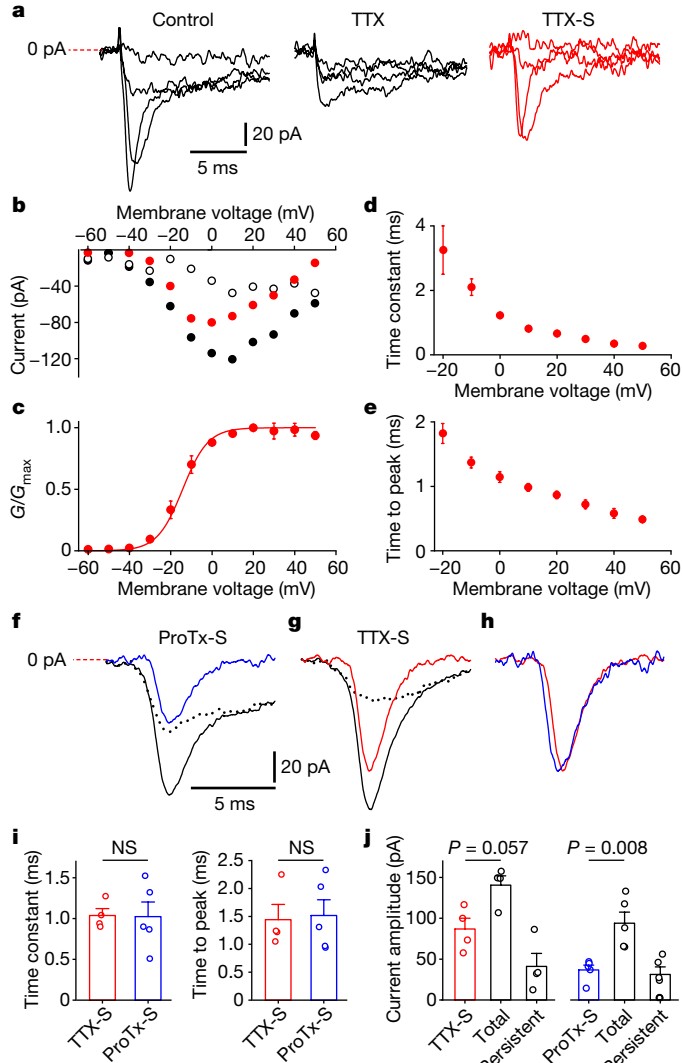

**Fig. 1 | TTX-S currents are present in OA chondrocytes and are produced largely by $Na_v1.7$ ProTx II-S channels. a**, Representative current traces in control buffer (left), in the presence of 1 μM TTX (middle), and resulting traces of the TTX-S current (right). **b**, Current–voltage curves of peak current amplitudes in control solution (●) and in the presence of 1 μM TTX (○), and the I-V curve for the TTX-S current (●). **c**, $G/G_{max}$ of the TTX-S current (mean ± s.e.m., $n = 3$) fitted with the Boltzmann equation. **d**, Single-exponential time constants of TTX-S current inactivation (mean ± s.e.m., $n = 3$). **e**, Time to peak of the TTX-S current (mean ± s.e.m., $n = 3$). **f**, Inhibition of fast-inactivating sodium current by 20 nM ProTx II. Averaged sodium current traces at 0 mV in control (black solid line) and in 20 nM ProTx II (black dotted line), and the resulting trace of their difference (ProTx II-S current, blue line). **g**, Inhibition of fast-inactivating sodium current by 1 μM TTX. Averaged traces of sodium currents evoked by 0 mV test voltage from −90 mV holding voltage in control solution (black solid line) and in the presence of 1 μM TTX (black dotted line), and the trace of their difference (TTX-S current, red trace). **h**, Overlays of TTX-S (red) and ProTx II-S (blue) current traces from **f,g**, normalized by peak current amplitudes. **i**, Left, inactivation time constants (mean ± s.e.m.) of TTX-S ($n = 4$) versus ProTx II-S ($n = 5$) currents. Right, time to peak of TTX-S (mean ± s.e.m., $n = 4$) versus ProTx II-S (mean ± s.e.m., $n = 5$) currents at 0 mV test voltage. **j**, Left, effect of 1 μM TTX on peak current amplitudes (mean ± s.e.m., $n = 4$) measured at 0 mV test voltage. Right, effect of 20 nM ProTx II on peak current amplitudes (mean ± s.e.m., $n = 5$) measured at 0 mV test voltage; averages of current amplitudes are shown for total, ProTx II-S and persistent sodium currents. $n$ indicates cell number; $P$ values by two-tailed Mann–Whitney test. NS, not significant.

was calculated from current–voltage data, averaged at the respective membrane voltages ($n = 3$), and fitted by Boltzmann equation with voltage for half-maximal activation ($V_{1/2}$) = −14.2 ± 0.8 mV, slope coefficient ($k$) = 6.1 ± 0.2 mV ($n = 3$) (Fig. 1c). On kinetic analysis, the falling phases of TTX-S current traces were best fitted with a single exponential. Inactivation time constants (single-exponential fits of the falling phase) were voltage-dependent, gradually decreasing with depolarizing membrane voltage from 3.3 ± 0.8 ms at −20 mV to 0.3 ± 0.1 ms ($n = 3$) at 50 mV membrane voltage (Fig. 1d). Time to peak of TTX-S current was measured from the voltage step onset and gradually decreased from 1.8 ± 0.2 ms at −20 mV to 0.5 ± 0.1 ms ($n = 3$) at 50 mV membrane voltage (Fig. 1e).

To further establish that the TTX-S currents included a $Na_v1.7$ component, we compared the time course of the TTX-S and ProTx II-sensitive (ProTx II-S) currents. Inhibition by 1 µM TTX was observed in 4 out of 4 cells studied. Inhibition by 20 nM ProTx II, which selectively blocks $Na_v1.7$, was seen in 5 out of 5 cells studied, confirming the presence of $Na_v1.7$.

To assess the effect of ProTx II, we recorded sodium currents before (Fig. 1f, black solid trace) and after (Fig. 1f, black dotted trace) application of 20 nM ProTx II; each trace is an average of 15–30 consecutive sweeps with 2–3 s inter-sweep interval, and ProTx II-S current (Fig. 1f, blue) was obtained by point-by-point subtraction of current in the presence of ProTx II from the current recorded in control. Similarly, we obtained current sensitive to 1 µM TTX (Fig. 1g, red trace). Supporting the notion that the TTX-S currents were largely produced by $Na_v1.7$, overlays of TTX-S (Fig. 1h, red trace) and ProTx II-S (Fig. 1h, blue trace) current traces normalized by peak current amplitudes showed similar time courses. Consistent with our observation of the similarities of TTX-S and ProTx II-S current time courses, their kinetic parameters were not significantly different. Inactivation time constants, obtained from a single-exponential fit of TTX-S and ProTx II-S currents elicited by 0 mV test voltage from −90 mV holding potential, were 1.0 ± 0.1 ms ($n = 4$) and 1.0 ± 0.2 ms ($n = 5$) (Fig. 1i), respectively, and their difference was not statistically significant ($P > 0.05$). Time-to-peak values of TTX-S and ProTx II-S currents at 0 mV membrane voltage were 1.4 ± 0.3 ms ($n = 4$) and 1.5 ± 0.3 ms ($n = 5$) (Fig. 1i), respectively, and were not statistically different ($P > 0.05$). These results add to the evidence that $Na_v1.7$ contributes a substantial proportion of the $Na^+$ current in OA chondrocytes. Total sodium current (average amplitude 141 ± 11 pA; $n = 4$) was inhibited by 1 µM TTX by 61.8 ± 16.4% ($n = 4$); TTX-S current amplitude at 0 mV was 87 ± 13 pA ($n = 4$) (Fig. 1j); and persistent (slow-inactivating, measured from control traces at 10 ms of test pulse onset) current amplitude was 41 ± 16 pA ($n = 4$). Total sodium current (93 ± 14 pA) was reduced by 20 nM ProTx II by 39.5 ± 13.5% ($n = 5$); ProTx II-S current amplitude at 0 mV was 37 ± 6 pA ($n = 5$); and persistent (slow-inactivating, measured from traces in control at 10 ms of test pulse onset) current amplitude was 31 ± 9 pA ($n = 5$) (Fig. 1j).

Together, these results establish the presence of $Na_v1.7$ in OA chondrocytes by demonstrating that $Na^+$ currents in these cells were sensitive to TTX, gating and kinetic properties similar to those known for $Na_v1.7$, and inhibition by 20 nM ProTx II, a selective $Na_v1.7$ blocker. The percentage of cells expressing $Na_v1.7$ can be seen from our patch clamp recordings, which were obtained from 77 human OA chondrocytes. Inward sodium currents with a fast-inactivating (millisecond time constant at 0 mV test voltage; Fig. 1d) component and peak amplitude above 30 pA were evoked in 17% of cells. Average inward current amplitude at 0 mV test potential in these cells was 82 ± 19 pA ($n = 13$), and average current density was 2.4 ± 0.4 pA pF⁻¹ ($n = 13$). A summary of the sodium current in human chondrocytes from three patients with OA is presented in Supplementary Table 1.

## $Na_v1.7$ deletion in chondrocytes protects from OA

To examine the role of $Na_v1.7$ in chondrocytes in OA and determine the relative contribution of DRG- and chondrocyte-expressed $Na_v1.7$ to OA

progression and pain, we generated mice with $Na_v1.7$ knockout in DRG neurons (hereafter referred to as *Nav1.7*[DRG]; $Na_v1.7$ is encoded by *Scn9a*), chondrocytes (hereafter referred to as *Nav1.7*[chondrocyte]), and both DRG neurons and chondrocytes (hereafter referred to as *Nav1.7*[DRG;chondrocyte]) by crossing $Na_v1.7$-floxed (*Nav1.7*[flox]) mice with *Nav1.8-cre* (*Nav1.8* is encoded by *Scn10a*) mice and/or *Agc1-cre*[ERT2] (*Agc1* is also known as *Acan*) mice, in which Cre-mediated recombination is induced by tamoxifen (Extended Data Fig. 2a). We confirmed $Na_v1.7$ deletions in DRG and chondrocyte following tamoxifen administration in adult mice (Extended Data Fig. 2b–d).

The two most common methods to experimentally model OA in mice include surgical and chemical induction. As in humans, both surgically and chemically induced mouse OA models exhibit articular cartilage erosion or loss and OA-related pain[23,24]. We established both surgically induced destabilization of the medial meniscus (DMM) (Fig. 2) and chemically induced MIA models (Extended Data Fig. 3a) in *Nav1.7*[flox] and *Nav1.7*[DRG;chondrocyte] mice. Histological analysis revealed that $Na_v1.7$ deletion in both DRG neurons and chondrocytes substantially attenuated cartilage loss, and reduced the Osteoarthritis Research Society International (OARSI) score (Fig. 2a,b and Extended Data Fig. 3b,c) in both DMM and MIA OA models. Notably, *Nav1.7*[DRG;chondrocyte] mice exhibited markedly reduced osteophyte formation, thickening of subchondral bone plate, suggestive of sclerosis, and decreased synovitis score in the DMM model (Fig. 2c–e). We evaluated the association of $Na_v1.7$ with OA pain by measuring open field movement activity and mechanical allodynia with von Frey testing. *Nav1.7*[DRG;chondrocyte] mice exhibited greater overall distance of movement and significantly reduced mechanical allodynia throughout the three-month period after DMM surgery and the four-week period after MIA injection relative to *Nav1.7*[flox] mice (Fig. 2f–h and Extended Data Fig. 3d–f). Immunohistochemical staining of knee joints indicated that OA-associated loss of the anabolic marker type II collagen (COL2), and increase of MMP13, aggrecan neoepitope generated via cleavage by ADAMTS5, and COMP fragment observed in *Nav1.7*[flox] mice, were inhibited in *Nav1.7*[DRG;chondrocyte] mice in both surgically and chemically induced OA (Extended Data Fig. 3g,h). These findings indicate that chondrocyte- and DRG neuron- expressed $Na_v1.7$ concurrently contribute to modulating OA progression and OA-associated pain.

To distinguish the contributions of $Na_v1.7$ expressed by chondrocytes and DRG neurons towards progression of OA pathology and OA-associated pain, we established the DMM OA model in *Nav1.7*[flox] and *Nav1.7*[chondrocyte] or *Nav1.7*[DRG] mice. Similar to our observations in *Nav1.7*[DRG;chondrocyte] DMM mice, *Nav1.7*[chondrocyte] mice also displayed substantial reductions in DMM-induced cartilage loss with reduced osteophyte formation, subchondral bone plate thickening and synovitis score compared with *Nav1.7*[flox] mice (Fig. 2i–m). Deletion of $Na_v1.7$ in chondrocytes reduced the loss of open field movement activity and mechanical allodynia (von Frey test) in the DMM model (Fig. 2n–p). In addition, deletion of $Na_v1.7$ in chondrocytes increased the amount of the anabolic effector COL2 and simultaneously decreased the amount of DMM-induced catabolic effectors, including MMP13, aggrecan neoepitope and COMP fragment (Extended Data Fig. 4a,b). By contrast, deletion of $Na_v1.7$ in DRG neurons reduced DMM-induced pain without affecting structural abnormalities such as cartilage destruction, osteophyte formation, subchondral bone plate thickening and synovitis score (Extended Data Fig. 4c–j). Immunohistochemical staining revealed that there was no difference between *Nav1.7*[flox] and *Nav1.7*[DRG] DMM mice in terms of the amounts of COL2, MMP13, aggrecan neoepitope and COMP fragment in cartilage (Extended Data Fig. 4k,l), supporting the notion that $Na_v1.7$ deletion in DRG neurons is not involved in the regulation of cartilage loss. In sum, $Na_v1.7$ expressed in chondrocytes modulates OA progression and resultant OA-associated pain behaviour, whereas $Na_v1.7$ expressed by DRG neurons contributes to OA-associated pain without affecting OA progression.

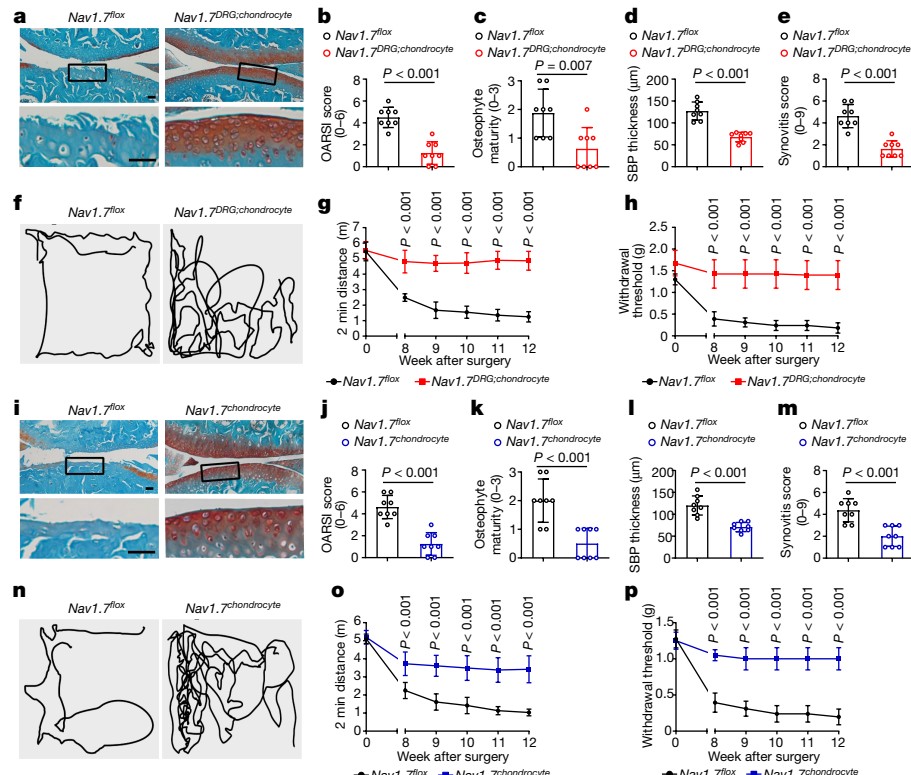

**Fig. 2 | Ablation of chondrocyte Na$_v$1.7 protects against OA and reduces pain. a**, Safranin O and Fast Green-stained sections of knee joints of mice with the indicated genotype ($n$ = 8). Scale bar, 50 μm. **b–e**, OARSI score (**b**), osteophyte development (**c**), subchondral bone plate (SBP) thickness (**d**) and synovitis score (**e**) in indicated mice 12 weeks after DMM ($n$ = 8). **f**, Traces of open field testing at 12 weeks after DMM surgery. **g,h**, Two-minute travel distance (**g**) and von Frey testing (**h**) in DMM-operated mice at the indicated time points after surgery ($n$ = 8). **i**, Safranin O and Fast Green-stained sections of knee joints ($n$ = 8). Scale bars, 50 μm. **j–m**, OARSI score (**j**), osteophyte development (**k**), SBP thickness (**l**), and synovitis score (**m**) in indicated mice 12 weeks after DMM surgery ($n$ = 8). **n**, Traces of open field testing at 12 weeks after DMM surgery. **o,p**, Two-minute travel distance (**o**) and von Frey testing (**p**) in DMM-operated mice at the indicated time points after surgery ($n$ = 8). **b–e,j–m**, Data are mean ± s.d., $P$ values by two-tailed unpaired Student's $t$-test. **g,h,o,p**, Data are mean ± 95% confidence interval (CI), $P$ values by two-tailed multiple unpaired Student's $t$-test with Welch's correction.

## Na$_v$1.7 inhibition protects against OA

We then sought to determine whether pharmacological Na$_v$1.7 blockade with PF-04856264, a selective Na$_v$1.7 blocker[25], could inhibit cartilage destruction and alleviate pain. To this end, we first established the DMM OA model in 12-week-old C57BL/6 wild-type male mice, followed by intra-articular injection of vehicle or PF-04856264 at 1.5 μg per g body weight every other day beginning 4 weeks post-operatively for a total of 8 weeks (Extended Data Fig. 5a). Safranin O staining revealed that PF-04856264 treatment protected the structure of articular cartilage and maintained proteoglycan content in the cartilage compared with vehicle (Extended Data Fig. 5b). PF-04856264 treatment was associated with significantly lower OARSI scores, osteophyte development and subchondral bone plate thickness relative to vehicle treatment (Extended Data Fig. 5c–e). In addition, DMM-induced OA pain behaviour was reduced COL2 level was increased in mice treated with PF-04856264 (Extended Data Fig. 5f–h). Accordingly, Col2 level was increased; by contrast, MMP13 and aggrecan neoepitope levels were significantly reduced in the cartilage of mice with DMM OA that received intra-articular injections of PF-04856264 compared with those treated with vehicle (Extended Data Fig. 5h,i). To explore potential sex differences, we replicated the DMM OA model in female wild-type mice using the same treatment approach. Consistent with the observations in male mice, female mice treated with PF-04856264 exhibited less articular cartilage loss, lower OARSI score and significant reductions in pain behaviour compared with untreated controls (Extended Data Fig. 5j–o).

To further demonstrate the effectiveness of Na$_v$1.7 blockade in preventing OA, we established the MIA model in wild-type male and female mice (Extended Data Fig. 6a). Since the route of drug delivery largely determines the drug concentrations achieved at the target site and thus therapeutic efficacy, and because systemic delivery is well accepted by patients, we determined the therapeutic effects of systemic administration of PF-04856264 through oral gavage at 30 μg per g body weight in the MIA model (Extended Data Fig. 6a). Similar to the results from local delivery, systemic delivery of PF-04856264 also markedly slowed OA progression as demonstrated by less cartilage loss, and improved OARSI score compared to vehicle treatment (Extended Data Fig. 6b–d). Furthermore, systemic PF-04856264 substantially increased the distance of movement in the open field test and von Trey hindpaw withdrawal threshold (Extended Data Fig. 6e,f). No apparent sex differences were observed, as PF-04856264 elicited cartilage loss protection and pain relief in both male and female mice (Extended Data Fig. 6b–f). Immunohistochemical staining revealed that oral PF-04856264 delivery increased COL2 expression and decreased MMP13 and aggrecan neoepitope levels in cartilage compared with vehicle (Extended Data Fig. 6g,h). Oral delivery of PF-04856264 also reduced cartilage loss in mice with DMM-induced OA (Extended Data Fig. 6i,j).

## CBZ reduces cartilage loss and pain in OA

Carbamazepine (CBZ) is a clinically used Na channel inhibitor that is known to act on Na$_v$1.7 (ref. 26) and has been approved by the US Food and Drug Administration (FDA) for indications in epilepsy, bipolar

disorder and neuropathic pain. To determine whether this sodium channel inhibitor has therapeutic effects on OA, we systemically delivered CBZ into MIA mice by oral gavage (250 mg per kg body weight, daily)—equivalent to the dosage used to treat epilepsy in humans, taking species differences into account[27], and suggested to alleviate disease severity in a metaphyseal chondrodysplasia Schmid type mouse model[28] (Extended Data Fig. 7a). Systemic CBZ treatment closely recapitulated the therapeutic effects of systemic PF-04856264 treatment on OA in terms of attenuation of cartilage loss, alleviation of OA pain and protection against changes in pro-anabolic and anti-catabolic effects in chondrocytes by upregulating COL2 expression and downregulating the expression of matrix-degrading enzymes (Extended Data Fig. 7b–h). We also evaluated the therapeutic effects of varying doses of CBZ administered orally in the DMM model (Extended Data Fig. 7i). In mice that underwent DMM surgery, a low dose of CBZ (10 mg per kg body weight) led to a significant reduction in cartilage loss 12 weeks after surgery. However, there was no significant difference in pain behaviour compared with untreated DMM mice (Extended Data Fig. 7j–m). Medium (50 mg per kg body weight) and high (250 mg per kg body weight) doses of CBZ provided stronger protection against cartilage loss and significantly reduced OA-associated pain behaviour compared with the low dose (Extended Data Fig. 7j–m). These findings suggest that non-specific VGSC blockers such as CBZ may hold promise as novel disease-modifying treatments for OA, and do not simply act in an analgesic manner.

## Na$_v$1.7 blockade regulates chondrocyte biology

Given that Na$_v$1.7 blockade led to pro-anabolic and anti-catabolic effects on chondrocytes in OA models, we next sought to determine whether Na$_v$1.7 blockade would affect chondrocyte biology in vitro. Expression of genes encoding catabolic molecules induced by IL-1β, including *MMP13*, *ADAMTS5*, *COX2* (also known as *PTGS2*) and *NOS2*, were inhibited by 1 µM TTX (Extended Data Fig. 8a–d). To further demonstrate the importance of Na$_v$1.7 in chondrocyte biology, we used Na$_v$1.7-selective blockers and a loss-of-function Na$_v$1.7 mutant approach. We used ProTx II[29] (half-maximal inhibitory concentration (IC$_{50}$) = 0.3 nM) and PF-04856264 (ref. 25) (IC$_{50}$ = 28 nM) to inhibit Na$_v$1.7 pharmacologically. Similar to TTX, blockade of Na$_v$1.7 with 25 nM ProTx II or 1 µM PF-04856264 (Extended Data Fig. 8a–d) demonstrated the importance of Na$_v$1.7 in inhibiting IL-1β-induced catabolism. To further assess whether Na$_v$1.7 blockade effectively inhibits chondrocyte catabolism in an inflammatory OA environment in general, we evaluated the effects of these inhibitors on chondrocyte catabolism in the presence of the additional inflammatory stimuli TNF[30] and polyinosinic–polycytidilic acid[31] (poly(I:C)). Simultaneous treatment of Na$_v$1.7 blockers significantly inhibited TNF- and poly(I:C)-induced chondrocyte catabolism (Extended Data Fig. 8e–l). VGSC blockade with TTX, ProTx II or PF-04856264 also significantly induced the expressions of genes encoding anabolic molecules such as *COL2* and *ACAN* (Extended Data Fig. 8m,n). We also isolated chondrocytes from *Nav1.7*$^{flox}$ and *Nav1.7*$^{chondrocyte}$ mice, and found that Na$_v$1.7 deletion blocked IL-1β-induced catabolism. Notably, Na$_v$1.7 deletion did not change anabolism under physiological conditions (Extended Data Fig. 8o–t).

To establish the relevance of our findings in a human context, we characterized the effects of Na$_v$1.7 on primary human chondrocytes isolated from six patients with KL 3–4 knee OA. We confirmed that both ProTx II and PF-04856264 decreased IL-1β-induced expression of *MMP13*, *ADAMTS5*, *COX2* and *NOS2* (Extended Data Fig. 9a–d) and increased the expression of *COL2* and *ACAN* (Extended Data Fig. 9e,f). To further determine the effects of Na$_v$1.7 blockers on human OA cartilage, we used an ex vivo cartilage explant assay using full-thickness OA cartilage from tibia plateaus of patients with OA who were undergoing total knee arthroplasty. In line with the results obtained from in vitro monolayer culture, supernatants of explants treated with Na$_v$1.7 blockers contained significantly lower levels of the cartilage catabolic marker MMP13 and higher levels of cartilage matrix component lubricin[32] (also known as proteoglycan 4 (PRG4)) compared with vehicle-treated controls (Extended Data Fig. 9g,h). In addition, Na$_v$1.7 blockers decreased the expression of catabolic markers *MMP13* and *ADAMTS5*, and increased the expression of anabolic markers *COL2* and *ACAN* in human OA cartilage explants cultured under inflammatory conditions (Extended Data Fig. 9i–l).

VGSCs are transmembrane proteins that open when the membrane potential in their vicinity becomes depolarized. There is increasing evidence that dynamic membrane potential influences a wide range of biological functions in both excitable and non-excitable cells; regulated secretion is one such function that has been widely studied[33,34]. In light of these observations, we hypothesized that alterations in secretion or cross-membrane transport of proteins might contribute to the regulation of chondrocyte biology induced by Na$_v$1.7 inhibition. We examined whether incubation with conditioned medium collected from chondrocytes treated with Na$_v$1.7-selective inhibitors induced the same regulatory effects on chondrocyte biology observed with Na$_v$1.7 inhibition. We found that conditioned medium collected from chondrocytes treated with PF-04856264 or ProTx II promoted the expression of anabolic markers and abolished IL-1β-induced expression of catabolic molecules, similar to direct treatment of chondrocytes with these Na$_v$1.7 inhibitors (Extended Data Fig. 9m–r). These findings suggest that secreted molecules in conditioned medium are responsible for the effects of Na$_v$1.7 blockade on chondrocyte biology.

We next attempted to isolate and characterize the molecules in conditioned medium that mediate Na$_v$1.7 regulation of chondrocyte anabolism and catabolism. To this end, we separated the components of conditioned medium into 4 fractions based on sized exclusion: <10 kDa, 10–30 kDa, 30–100 kDa and over 100 kDa, and tested their effects on chondrocyte biology (Extended Data Fig. 10a). We found that the 30–100 kDa fraction was capable of promoting chondrocyte anabolism without affecting catabolism, whereas the 10–30 kDa fraction only inhibited cytokine-induced catabolism (Extended Data Fig. 10b–e). These findings suggest that one or more secreted proteins in the 30–100 kDa and the 10–30 kDa molecular weight range in conditioned medium from cells treated with selective Na$_v$1.7 blockers are responsible for enhancing anabolism and inhibiting catabolism, respectively.

We analysed these two fractions of conditioned medium using tandem mass spectrometry (MS/MS). The MS/MS spectra were searched against the UniProt database, using Sequest within Proteome Discoverer. We defined candidate targets on the basis of the following properties: (1) having a molecular weight between 10 and 30 kDa or between 30 and 100 kDa; (2) exhibiting unique or more than twofold increased expression after treatment with both PF-04856264 and ProTx II relative to vehicle treatment; and (3) being secreted proteins. Of the six hits that met these criteria in the 10–30 kDa and 30–100 kDa fractions of both PF-04856264- and ProTx II-treated conditioned medium (Fig. 3a,b), we focused on HSP70 (encoded by *HSPA1A* and *HSPA1B*) and midkine (encoded by *MDK*) because these two proteins were previously reported to be implicated in chondrocyte biology, and over-expression of *HSP70* (ref. 35) and treatment with recombinant midkine[36] have been reported to protect against OA. After treatment with selective Na$_v$1.7 blockers and the pan-Na$_v$ blocker CBZ, which is known to act on Na$_v$1.7 (ref. 26), both HSP70 and midkine were significantly upregulated in the medium, even though the amount of HSP70 in the cell lysate remained unchanged. Midkine was also upregulated in the cell lysate (Extended Data Fig. 11a–d), suggesting that Na$_v$1.7 blockers increase medium HSP70 and midkine by modulating the chondrocyte secretion. To further validate the biological regulation of HSP70 and midkine, we treated chondrocytes with the recombinant proteins. HSP70 dose-dependently enhanced chondrocyte anabolism without

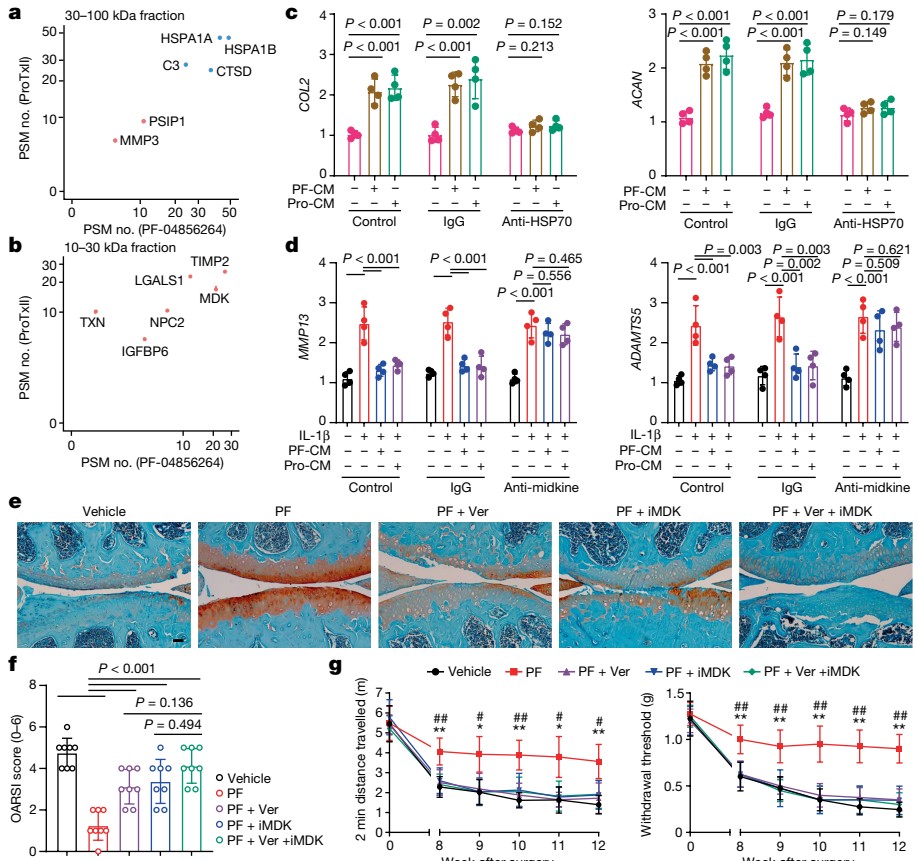

**Fig. 3 | Blockade of Na$_v$1.7 regulates chondrocyte biology through enhancing HSP70 and midkine secretion. a**, Peptide-spectrum match (PSM)-based abundance of proteins identified from the 30–100 kDa fraction of conditioned medium that are unique to (red dots) or increased with (blue dots) PF-04856264 and ProTx II treatment. **b**, PSM-based abundance of proteins (red dot) identified from the 10–30 kDa fraction of conditioned medium that are unique to PF-04856264 and ProTx II treatment. **c**, *COL2* and *ACAN* mRNA levels in human C28I2 chondrocytes stimulated with conditioned medium collected from cells treated with ProTx II (Pro-CM) or PF-04856264 (PF-CM) in the absence or presence of IgG or anti-HSP70 antibodies ($n = 4$ biological replicates). **d**, *MMP13* and *ADAMTS5* mRNA levels in C28I2 chondrocytes stimulated with IL-1β and conditioned medium collected from cells treated with ProTx II (Pro-CM) or PF-04856264 (PF-CM) in the absence or presence of control IgG or anti-midkine antibodies ($n = 4$ biological replicates). **e**, Safranin O and Fast Green-stained knee joint sections ($n = 8$). Ver, VER 155008. Scale bar, 50 μm. **f**, OARSI score from images represented in **e**. **g**, Two-minute travel distance and von Frey testing at the indicated time points with indicated treatments after DMM surgery ($n = 8$). Data are mean ± 95% confidence interval, $P$ values by two-way ANOVA with Bonferroni post hoc test. *Vehicle versus PF, $P < 0.05$; **vehicle versus PF, $P < 0.01$; #PF versus PF + Ver + iMDK, $P < 0.05$; ##PF versus PF + Ver + iMDK, $P < 0.01$. **c,d,f**, Data are mean ± s.d., $P$ values by one way ANOVA with Bonferroni post hoc test.

affecting catabolism (Extended Data Fig. 11e,f), whereas midkine dose-dependently inhibited IL-1β-induced chondrocyte catabolism without having an effect on anabolism (Extended Data Fig. 11g,h). Notably, specific blockade of HSP70 and midkine in conditioned medium using antibodies abolished the effects of conditioned medium on chondrocyte biology (Fig. 3c,d).

Given the finding that Na$_v$1.7 blockers regulate chondrocyte biology by increasing secretion of HSP70 and midkine in vitro, we tested whether blocking HSP70 and/or midkine affected protective effects against OA mediated by Na$_v$1.7 inhibition in vivo. Administration of the Na$_v$1.7 blocker PF-04856264 resulted in a reduction in cartilage loss and a decrease of OA-associated pain behaviour in DMM mice (Fig. 3e–g). However, this protective effect was nearly abolished by combined application of the HSP70 inhibitor VER 155008 and the midkine inhibitor iMDK (Fig. 3e–g). Of note, VER 155008 or iMDK alone also significantly reduced the protective effects of PF-04856264 against OA without significantly affecting OA-associated pain (Fig. 3e–g). Collectively, these findings underscore the importance of HSP70 and midkine in maintaining the protective effects of Na$_v$1.7 inhibition against OA in vivo.

Consistent with our observations that Na$_v$1.7 blockers upregulate medium HSP70 and midkine in human chondrocytes, we found that genetic deletion of Na$_v$1.7 increased the secretion of HSP70 and

midkine in mouse chondrocytes isolated from DMM mice (Extended Data Fig. 11i,j). In addition, serum levels of HSP70 and midkine were markedly increased in DMM mice compared with sham surgery mice (Extended Data Fig. 11k,l). To verify the clinical relevance of our findings, we measured the levels of HSP70 and midkine in serum and synovial fluid from healthy individuals and patients with symptomatic knee OA (Supplementary Tables 2 and 3). Serum HSP70 and midkine levels were significantly higher in patients with OA than in healthy controls (Extended Data Fig. 11m,o). For a subset of patients with symptomatic knee OA for whom both serum and synovial fluid samples were available, serum levels of HSP70 and midkine positively correlated with synovial fluid levels (Extended Data Fig. 11n,p). Collectively, these results reveal that Na$_v$1.7 blockade regulates chondrocyte anabolism and catabolism at least in part by regulating the secretions of HSP70 and midkine, respectively.

## Na$_v$1.7 blockade alters chondrocyte Ca$^{2+}$

Previous studies have shown that VGSCs contribute to the regulation of intracellular Ca$^{2+}$ signalling in non-excitable glial cells[37]. Ca$^{2+}$ serves as a crucial second messenger, with a pivotal role in dynamic regulation of diverse cellular processes, including protein secretion[38]. Here

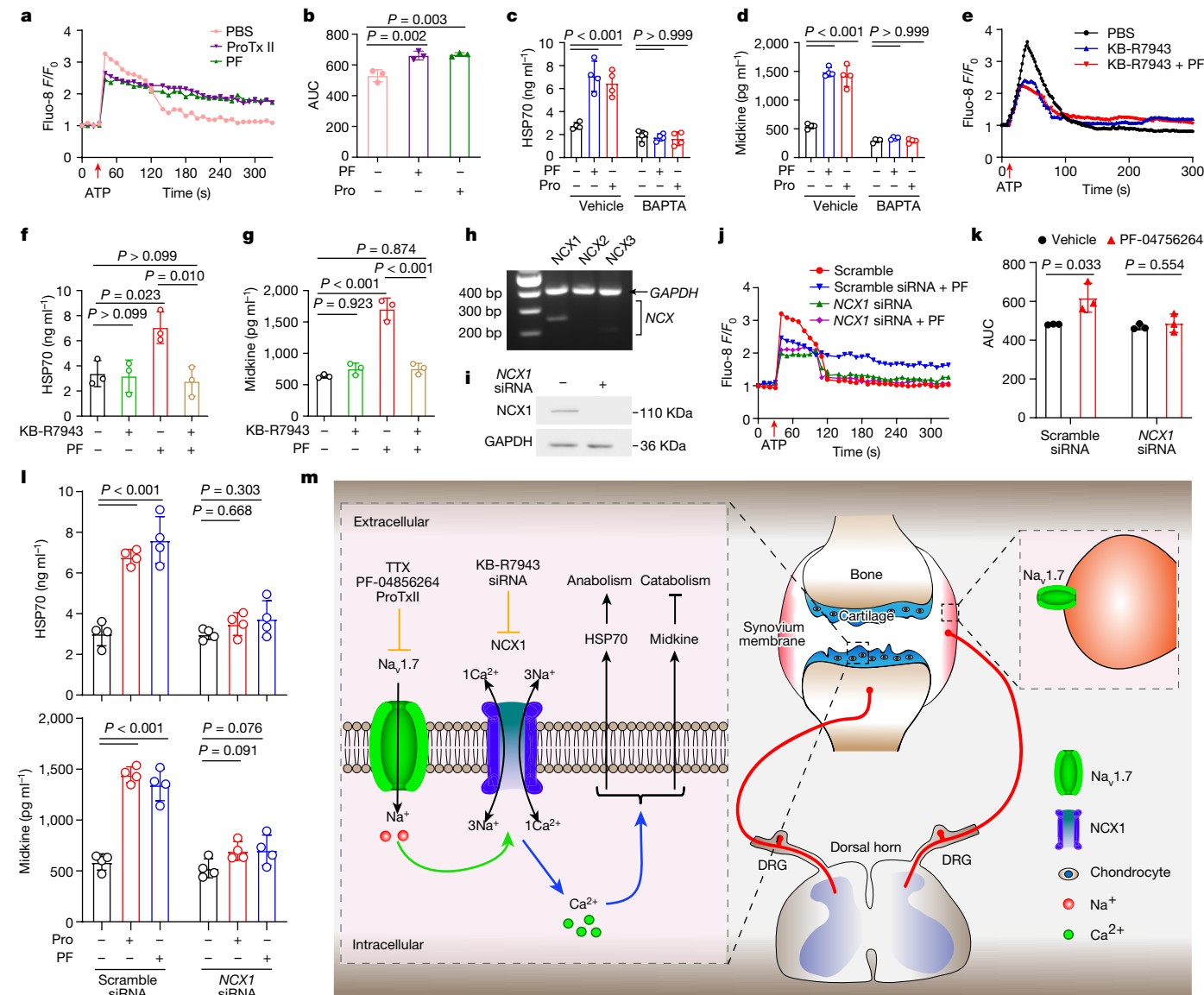

**Fig. 4 | Ca²⁺ signalling in chondrocytes. a,b**, $F/F_0$ (**a**) and area under the curve (AUC) of intracellular $Ca^{2+}$ (**b**) in human OA chondrocytes following ATP stimulation, measured by plate reader. ATP present from red arrow. **c,d**, HSP70 (**c**) and midkine (**d**) levels in conditioned medium of C28I2 cells pre-treated with BAPTA-AM, followed by ProTx II or PF-04856264. **e**, $F/F_0$ of intracellular $Ca^{2+}$ in KB-R7943 treated C28I2 chondrocytes following ATP stimulation, assayed by confocal fluorescence microscopy. **f,g**, HSP70 (**f**) and midkine (**g**) levels in conditioned medium of C28I2 cells treated with KB-R7943 in the presence or absence of PF-04856264. **h**, Expression of NCX isoforms in C28I2 cells. **i**, Knockdown efficiency of NCX1 in C28I2 cells. **j,k**, $F/F_0$ (**j**) and AUC of intracellular $Ca^{2+}$ (**k**) following ATP stimulation in C28I2 chondrocytes transfected with scramble or *NCX1* siRNA measured by plate reader. **l**, HSP70 and midkine levels in conditioned medium of chondrocytes transfected with scramble or *NCX1* siRNA and treated with ProTx II or PF-04856264. **m**, Model of mechanisms of chondrocyte- and DRG-expressed Na$_v$1.7 in OA, and amelioration of OA and pain via Na$_v$1.7 blockade. **b–d,f,g,k,l**, Data are mean ± s.d. **b,f,g**, $P$ values calculated by one way ANOVA with Bonferroni post hoc test. **k**, Two-tailed unpaired Student's $t$-test. **c,d,l**, Two-way ANOVA with Bonferroni post hoc test. **b,f–i,k**, $n = 3$ biological replicates. **c,d,l**, $n = 4$ biological replicates.

we investigated the role of Na$_v$1.7 blockade-mediated intracellular $Ca^{2+}$ signalling in the regulation of HSP70 and midkine secretion in chondrocytes. Initially, we assessed the effect of Na$_v$1.7 inhibition on the ATP-triggered increase in intracellular Na$^+$ in human OA chondrocytes and C28I2 chondrocytes. Inhibition of Na$_v$1.7 with PF-04856264 attenuated the increase in intracellular Na$^+$ (Extended Data Fig. 12a,b). Inhibition of Na$_v$1.7 with ProTx II or PF-04856264 was followed by an increase in intracellular $Ca^{2+}$ levels in both human OA chondrocytes and C28I2 cells. Specifically, Na$_v$1.7 blockade decreased the initial intracellular $Ca^{2+}$ surge triggered by ATP stimulation within approximately 100 s. Subsequently, Na$_v$1.7 blockade resulted in higher sustained $Ca^{2+}$ levels than in control cells. These observations were confirmed

using confocal microscopy and plate reader measurements (Fig. 4a,b and Extended Data Fig. 12c–f). To validate the importance of intracellular $Ca^{2+}$ levels in the modulation of HSP70 and midkine secretion in chondrocytes, we conducted experiments on C28I2 cells using the $Ca^{2+}$ ionophore ionomycin and the cell-permeant $Ca^{2+}$ chelator BAPTA-AM. The results demonstrated that higher $Ca^{2+}$ signals contribute to the regulation of HSP70 and midkine secretion via Na$_v$1.7 blockade. Notably, there was a loss of the enhanced secretion of HSP70 and midkine by Na$_v$1.7 blockade in chondrocytes when BAPTA-AM was present (Fig. 4c,d and Extended Data Fig. 12g,h). We previously documented the involvement of Na$^+$/Ca$^{2+}$ exchange in the modulation of $Ca^{2+}$ signals by VGSC blockade in astrocytes and microglial cells[37,39].

To assess the role of NCX family proteins (also known as solute carrier family 8 proteins) in regulating intracellular calcium levels after $Na_v1.7$ blockade in chondrocytes, we used the pharmacological inhibitor KB-R7943 to inhibit NCX prior to ATP stimulation. These experiments showed that KB-R7943 significantly decreased the ATP-induced surge of intracellular $Ca^{2+}$ within the first 100 s. Additionally, when NCX was blocked with KB-R7943, PF-04856264 had minimal effect on $Ca^{2+}$ levels in chondrocytes. These results indicate that NCX contributes to the regulation of intracellular $Ca^{2+}$ signals by $Na_v1.7$ blockade (Fig. 4e). Consistent with these results, NCX inhibition abolished the enhanced secretion of HSP70 and midkine caused by $Na_v1.7$ blockade (Fig. 4f,g). We also used RT–PCR to examine the expression of *NCX1* (also known as *SLC8A1*), *NCX2* (*SLC8A2*) and NCX3 (*SLC8A3*) in C28I2 cells. This experiment demonstrated clear expression of *NCX1* mRNA in the chondrocytes, whereas *NCX2* and *NCX3* were undetectable (Fig. 4h). We then showed that PF-04856264-mediated regulation of $Ca^{2+}$ signals and secretion of HSP70 and midkine were essentially lost in chondrocytes following short interfering RNA (siRNA) knockdown of NCX1 (Fig. 4i–l). Collectively, these findings indicate that intracellular $Ca^{2+}$ signals are essential for the enhanced secretion of HSP70 and midkine following $Na_v1.7$ blockade. Moreover, these findings underscore the critical role of NCX1 in regulating calcium signalling and the associated protein secretion by $Na_v1.7$ blockade in chondrocytes.

## Discussion

Although classically considered as the substrate for action potential initiation and propagation[40], low densities of VGSCs have been reported in multiple cell types that have traditionally been considered as non-excitable, including macrophages, microglia and astrocytes, where they may contribute to the regulation of effector functions such as phagocytosis, motility, cytokine release and response to injury[41–43]. Chondrocytes express a variety of ion channel types[44,45] that participate in diverse physiological processes, including setting resting potential, mechanoresponsiveness, volume regulation, calcium signalling, bone development, intracellular pH regulation, cellular biosynthesis and proliferation[46–51]. Notably, the expression of several ion channel types is known to be altered in OA chondrocytes[44,45,52], whereas cartilage-specific knockout of mechanosensory ion channels decreases age-related OA[53].

Sugimoto et al. reported a TTX-S current in rabbit articular chondrocytes[2]; however, its molecular identity was unknown. Here we demonstrate that human OA chondrocytes express TTX-S sodium current that is produced mainly by functional $Na_v1.7$. We observed a sodium current density of 2.4 pA pF$^{-1}$ (39.7 pS pF$^{-1}$) in human chondrocytes. Assuming single-channel conductance[54] of 6.4 pS and open probability[10,54] of 0.4–0.6 at 0 mV, this suggests a channel density of 0.1–0.15 channels per µm$^2$ and 350–525 channels per cell. Notably, TTX-S currents in human OA chondrocytes comprise 62% of the total sodium current on average, whereas ProTx II-S currents contribute 40%. This result highlights the existence of $Na_v1.7$ sodium channels in primary human OA chondrocytes and points to the possibility of a fractional presence of non-$Na_v1.7$ TTX-S sodium channels in this cell type.

$Na_v1.7$ is known to regulate cytokine secretion in dendritic cells[42]. Here we have demonstrated that $Na_v1.7$ blockers affect chondrocyte anabolic and catabolic processes through regulation of the chondrocyte secretome. The combination of fractionation of conditioned medium and subsequent proteomics analysis led to the identification of HSP70 and midkine as key molecules in conditioned medium from cells treated with $Na_v1.7$ inhibitors that participate in the control of chondrocyte metabolism. Increased release of HSP70 and midkine by blockade of $Na_v1.7$ in $Na_v1.7$-expressing chondrocytes may have both autocrine and paracrine effects and enable $Na_v1.7$-expressing and non-expressing chondrocytes to integrate signalling from the local environment to orchestrate the anabolic and catabolic processes

that contribute to OA. We have demonstrated that increased intracellular $Ca^{2+}$ signals are essential for the enhanced secretion of HSP70 and midkine induced by $Na_v1.7$ blockade in chondrocytes. This effect was nullified by pharmacological inhibition and genetic ablation of NCX1, highlighting the crucial role of NCX1 in regulating intracellular $Ca^{2+}$ signalling and subsequent secretion of HSP70 and midkine in response to the $Na_v1.7$ blockade. Of note, $Na_v1.7$ blockade elicits a distinct sequence of intracellular calcium level changes in chondrocytes: it initially reduces $Ca^{2+}$ levels stimulated by ATP, which is counteracted by NCX1 inhibition; subsequently, there is a sustained elevation of $Ca^{2+}$ levels, potentially orchestrated by a series of protein–protein and protein–lipid interactions. Our results suggest that $Na_v1.7$ blockers hold promise as therapeutic agents to both protect against cartilage loss and attenuate pain in OA. We demonstrate in multiple animal models that CBZ, a sodium channel blocker currently in clinical use, prevents cartilage loss in animal models of OA, an effect beyond purely blocking pain perception. These results highlight the potential clinical application of a currently available, FDA-approved sodium channel blocker that might be repurposed for the treatment of OA.

In conclusion, we identify $Na_v1.7$ as an OA-associated ion channel with dual roles in pain and cartilage homeostasis, with DRG neuron-expressed $Na_v1.7$ being involved in pain and chondrocyte-expressed $Na_v1.7$ governing chondrocyte biology, cartilage loss and resultant pain in OA through a powerful effect on the chondrocyte secretome (Fig. 4m). Identification of $Na_v1.7$ as a novel chondrocyte-expressed, OA-associated gene uncovers a target for the development of therapies that may provide both disease-modifying and non-addictive pain relief treatment for OA.

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

## Methods

### Mice

C57BL/6 and *Agc1-cre^ERT2* mice were obtained from The Jackson Laboratory. *Nav1.8-cre;Nav1.7^flox/flox* mice were mated with transgenic mice expressing *Agc1-cre^ERT2* to obtain inducible Na$_v$1.7-knockout mice in chondrocytes and both chondrocytes and DRGs. For activation of *cre^ERT2* in adult mice, 150 mg kg$^{-1}$ body weight of tamoxifen (Sigma, T5648) in sunflower seed oil (Sigma, S5007) was injected intraperitoneally into 10-week-old mice once a day for 5 consecutive days. Littermate controls were used for all experiments. All animals were housed on a 12-h light:dark cycle with ad libitum access to food and water in a specific pathogen-free environment. Animals were maintained on a C57BL/6 J background, and age matched males typically at 12 weeks of age were used, unless otherwise specified in the figure legends. No statistical methods were used to predetermine sample size. All animal studies were performed in accordance with institutional guidelines and approved by the Institutional Animal Care and Use Committee of New York University Grossman School of Medicine.

To establish the surgically induced DMM[55] model, after ketamine and xylazine anaesthesia, the medial meniscotibial ligament in the right knee was sectioned with a blade to destabilize the medial meniscus. The chemically induced MIA OA model was established unilaterally via intra-articular injection of 0.1 mg of MIA (Sigma, I2513) in 6 µl of 0.9% sterile saline with a 30-gauge needle after anaesthetization with ketamine and xylazine[56]. Mice with OA models were randomized to receive different treatment within a cage. PF-04856264 (Alomone labs, 1235397-05-3), at 30 µg per g body weight or 1.5 µg per g body weight was orally delivered or injected intra-articularly, respectively, daily over a 4-week period starting from the first day of MIA injection. PF-04856264 was intra-articular injected at 1.5 µg per g body weight or orally delivered at 30 µg per g body weight every other day starting from 4 weeks post DMM surgery for a total of 8 weeks. CBZ (Sigma, C4024) was delivered through oral gavage at 250 mg per kg body weight daily over a 4-week period starting from the first day of MIA injection. Additionally, doses of 10, 50 or 250 mg per kg body weight CBZ were administered daily via oral gavage over an 8-week span, commencing 4 weeks after DMM surgery. To determine whether blocking HSP70 and/or midkine affected Na$_v$1.7 blocker PF-04856264's protective effects against OA in vivo, mice received oral administration of PF-04856264 at a dosage of 30 µg per g body weight every day, and they were simultaneously subjected to intra-articular injections of VER 155008 (Sigma, SML0271) at 0.55 µg/g body weight, iMDK (Tocris, 5126) at 0.9 µg per g, or a combined application of VER 155008 and iMDK daily starting from 4 weeks after DMM surgery for a total of 8 weeks.

### Behaviour tests

OA-associated pain was measured using the von Frey assay and the open field travel analysis[57] three times before establishment of the OA model, and every week starting from 8 weeks after DMM surgery, or at day 2, 4, 8, 14, 20 and 28 post MIA injection. All the behavioural tests were conducted in a blinded manner and performed between the hours of 12:00 and 17:00. von Frey filaments (Stoelting) were applied with increasing force intensities on the plantar surface of the hindpaw of the mouse which is placed in an elevated Plexiglass chamber with a metal grid floor that gave access to the plantar surface of the paws to determine the tactile pain threshold as based on a previous publication[57]. Rapid withdrawal of the hindpaw was recorded as a positive response. Hind paws were subjected to 10 trials at a given intensity with a 30-s interval maintained between trials and the number of positive responses for each von Frey filament's stimulus was recorded. Animals were considered to have reached tactile threshold when 5 out of 10 trials generated a positive response. For open field travel analysis, mice were placed individually in a square clear chamber (45 × 45 cm) and allowed to freely explore for 2 min under normal lighting. Movement and trajectories of the mice were videoed and analysed by a computerized system.

### Human subjects research

Human subjects research was performed according to the Institutional Review Boards at New York University Medical Center (institutional review board (IRB) study number i11-01488 and i9018). Human OA cartilage samples were collected from patients receiving total knee joint replacement surgery for OA at New York University Langone Orthopaedic Hospital. Non-arthritic femoral condyle cartilage specimens were obtained from fresh osteochondral allografts discarded following donor plug collection during surgical osteochondral allograft implantation. Cartilage samples used are surgical discards, and no consent is required based on our approved IRB, as we do not collect patient information except age, sex and clinical diagnosis of the samples, such as Osteoarthritis. OA and non-arthritic cartilage specimens were stored in liquid nitrogen immediately after collection until protein or RNA extraction.

A total of 22 non-OA and 165 patients with symptomatic knee OA from the New York biomarker cohort[58] were included in this study according to the American college of rheumatology (ACR) criteria. The demographic data are summarized in Supplementary Table 2. Informed consent was obtained from all participating subjects. The IRB of the New York University Grossman School of Medicine approved this study (no. i05-131).

OA synovial fluid and serum samples were collected as part of an observational study to determine factors influencing knee OA pain improvement with hyaluronic acid visco-supplementation[59]. The synovial fluid samples were collected without joint lavage, and the volume ranged from 0.5 to 30 ml. The cell-free synovial fluids were prepared and frozen (−80 °C) within 1 h of collection. Collection and storage of synovial fluids were approved (no. 13-01257) by the IRB of the NYU Grossman School of Medicine. The demographic data are summarized in Supplementary Table 3.

For the full-thickness cartilage explant assay, human tibia plateaus were obtained from 8 deidentified patients with OA undergoing total knee arthroplasty. For each individual patient with OA, 12 full-thickness cartilage explants were isolated from areas with various degrees of OA-related cartilage degeneration with a 3-mm biopsy punch and randomly distributed into three different groups and treated with 10 ng ml$^{-1}$ IL-1β, 10 ng ml$^{-1}$ IL-1β plus 25 nM ProTx II, or 10 ng ml$^{-1}$ IL-1β plus 1 µM PF-04856264 in DMEM medium for 5 days. The supernatant was collected and spun at 200*g* at 4 °C, followed by ELISA assay.

### Electrophysiology

Primary human chondrocytes from patients with OA were grown in 100 mm tissue culture dishes in growth medium DMEM (Gibco, 11995-065) supplemented with 10% FBS (Hyclone, SH30088.03) and 1× penicillin−streptomycin (Thermo Fisher, 15070063). Chondrocytes were passaged every 5−7 days at 75−80% confluency no more than 3 times. Cells were plated at passaging into 12-mm round glass poly-D-lysine/laminin-coated coverslips (Corning, 354087) in 24-well plate format according to the following protocol: growth medium was removed, chondrocytes were rinsed once with 5 ml Ca$^{2+}$ and Mg$^{2+}$-free DPBS (Gibco, 14190-144) and incubated for 3−5 min with 1.5 ml 0.25% Trypsin/EDTA (Corning, 10222017), then cells were gently lifted off the dish and pipette-triturated in 8.5 ml of growth medium. Twenty-five microlitres of homogenized chondrocytes suspension was diluted into 1 ml growth medium at each cover glass to reach optimal cell density and were maintained in growth medium for 3−6 days until electrophysiological recordings.

Currents were recorded in whole-cell voltage clamp by Axopatch 200B amplifier (Molecular Devices). Recordings were low-pass filtered at 2 kHz and acquired at 100 kHz by Digidata 1440 A DAC using

Clampex 10.7 software (Molecular Devices). p/4 leak subtraction protocol and sweep-averaging were used to subtract uncompensated leak and capacitance currents and to enhance signal/noise ratio. Pipettes were pulled from glass capillaries (PG52165-4; WPI) and had resistance 2–3.5 MΩ when filled with intracellular solution (in mM): 140 CsF, 10 NaCl, 10 HEPES, 1 EGTA, 20 dextrose, pH 7.3 with CsOH (328 mOsm l$^{-1}$ with sucrose). Extracellular solution contained (in mM): 145 NaCl, 4 KCl, 2 CaCl$_2$, 2 MgCl$_2$, 10 HEPES; 10 TEA-Cl, 10 Dextrose, pH 7.4 with NaOH (327 mOsm l$^{-1}$). Solutions for sodium current isolation were from ref. 10. The liquid junction potential was not compensated. Recordings were made at room temperature. Data were analysed using pClamp 10.7 (Molecular Devices) and Origin 2022b (OriginLab) software.

## RNA extraction from human cartilage

For RNA isolation, about 1 g of cartilage was pulverized in liquid nitrogen and homogenized in Trizol at a concentration of 1 g tissue per 10 ml Trizol (Invitrogen, 15596026), followed by incubation at 4 °C with rotating for 2 h. Samples were mixed with 0.2 volumes of chloroform, vortexed for 20 s, and centrifuged at 14,000 rpm for 20 min at 4 °C. The aqueous phase was collected and gently mixed with an equal volume of isopropanol, followed by centrifugation at 14,000 rpm for 20 min at 4 °C. The resulting pellet was suspended in 350 μl of RLT buffer and processed for cleanup using the RNeasy Mini Kit (Qiagen, 74104) following the manufacturer's instructions.

## RNA assay by RT–qPCR

Total RNA extracted from chondrocytes or human cartilage was reverse transcribed using the High-Capacity cDNA Reverse Transcription Kit (Applied Biosystems, 4387406). RT–qPCR was performed in triplicate with SYGR Green (Applied Biosystems, A25780) using human or mouse primers to *Acan*, *Col2*, *Mmp13*, *Adamts5*, *Cox2*, *Nos2* and *Gapdh* (Applied Biosystems Real-time PCR system). mRNA levels were normalized to *Gapdh* and reported as relative mRNA fold change.

## Histology

Human cartilage or mouse knee joints were fixed in 4% paraformaldehyde for 24 h before decalcification in 10% w/v EDTA for 2 weeks before paraffin embedding. The paraffin blocks were sectioned at a thickness of 5 μm and serial sections were subjected to Safranin O or haematoxylin and eosin (H&E) staining. Cartilage destruction was graded on Safranin O-stained sections by blinded observers using the OARSI histology scoring system[60] (grade 0–6). Osteophyte development[61] (grade 0–3) was evaluated and the thickness of the subchondral bone plate[62] was measured on Safranin O or H&E-stained sections. Synovitis (grade 0–9) was determined based on the synovial lining cell layer enlargement, resident cell density, and inflammatory infiltration on H&E-stained sections[63]. For immunohistochemical staining, deparaffinized and hydrated sections were incubated with 0.1% trypsin for 30 min at 37 °C, followed by 0.25 U ml$^{-1}$ chondroitinase ABC (Sigma-Aldrich, C3667) and 1 U ml$^{-1}$ hyaluronidase (Sigma-Aldrich, H3560) for 60 min at 37 °C, respectively. After blocking, the sections were incubated with antibodies against Na$_v$1.7 (1:50, Alomone Labs, ASC-008), COL2 (Invitrogen, cat. no. MA5-12789), COMP fragment[64] (1:200, affinity-purified monoclonal), aggrecan neoepitope (1:100, Millipore, AB8135) and MMP13 (1:200, Abcam, ab3208) overnight at 4 °C. Detection was performed using the Vectastain Elite ABC kit (Vector Laboratories, PK6100), and the positive signal was visualized with 0.5 mg ml$^{-1}$ 3,3-diaminobenzidine in 50 mM Tris-Cl substrate (Sigma-Aldrich, D12384) and then counterstained with 1% methyl green (Sigma-Aldrich, 67060). Images were acquired with a Zeiss microscope. Semi-quantification analysis of the density of immunohistochemical staining for Aggrecan neoepitope, COMP fragment and collagen X was performed by ImageJ, and the same signal threshold was used for each group of similar immunohistochemical images[65].

## Cell culture

Primary articular chondrocytes were isolated from the femoral condyles and tibial plateaus of *Nav1.7$^{flox}$* and *Nav1.7$^{chondrocyte}$* mice on postnatal day 6 (ref. 66). Chondrocytes were maintained as a monolayer in Dulbecco's modified Eagle's medium (DMEM) supplemented with 10% FBS, 50 U/ml penicillin, and 0.05 mg ml$^{-1}$ streptomycin. Articular chondrocytes at culture day 2 were treated as indicated for each experiment.

Primary articular chondrocytes were isolated from the femoral condyles and tibial plateaus of *Nav1.7$^{flox}$* and *Nav1.7$^{chondrocyte}$* mice at 12 weeks after DMM surgery. Chondrocytes were maintained as a monolayer in Dulbecco's modified Eagle's medium (DMEM) supplemented with 10% FBS, 50 U ml$^{-1}$ penicillin, and 0.05 mg ml$^{-1}$ streptomycin. After 5 days, the conditioned medium was collected and spun at 200$g$ at 4 °C, followed by ELISA assay.

Human C28I2 chondrocytes were grown in DMEM medium supplemented with 10% FBS, 50 U ml$^{-1}$ penicillin, and 0.05 mg ml$^{-1}$ streptomycin. To knockdown Na$_v$1.7, cells were transfected with commercially available siRNA (S534077, A134907) using Lipofectamine (Invitrogen, 13778100) as instructed by manufacture's protocol. To fraction conditioned medium, C28I2 cells were stimulated with or without 25 nM ProTx II or PF-04856264. In brief, C28I2 cells were cultured with DMEM medium supplemented with 10% FBS, 50 U ml$^{-1}$ penicillin and 0.05 mg ml$^{-1}$ streptomycin. When cells reached 80% confluency, the medium was changed to DMEM supplemented with ITS Liquid Media Supplement (Sigma-Aldrich, I3146) containing 25 nM ProTx II or PF-04856264 for 2 days. The conditioned medium was then collected. After centrifugation to get rid of the cell debris, the medium was then fractioned based on molecular weight to >100 kDa, 30–100 kDa, 10–30 kDa and <10 kDa using Amicon Ultra-2 Centrifugal Filter Units sequentially (Fisher Scientific, UFC210024, UFC203024 and UFC201024). PF-04856264 and ProTx II have molecular weight of 437.492 and 3,826.65 Da, respectively. To exclude the effects of PF-04856264 and ProTx II in conditioned medium, ProTx II was depleted using dialysis tubing that allows the removal of molecules with molecular weights between 3.5–5 kDa (Micro Float-A-Lyzer 3.5–5 kDa, F235053, Thomas Scientific), while PF-04856264 was simply removed through dialysis against the medium. Conditioned medium with molecular weight of 30–100 kDa and 10–30 kDa were then analysed by mass spectrometry, performed by NYU Proteomics Laboratory. All MS/MS spectra were collected using the following instrument parameters: resolution of 15,000, automatic gain control (AGC) target of 5e4, maximum ion time of 120 ms, one microscan, 2 $m/z$ isolation window, fixed first mass of 150 $m/z$, and normalized collision energy (NCE) of 27. MS/MS spectra were searched against a UniProt Human database using Sequest within Proteome Discoverer 1.4.

## Western blotting

Western blot analyses were conducted with protein lysates from primary human cartilage and C28I2 cells. To determine the membrane localization of Na$_v$1.7 in chondrocytes, cytosolic and membrane fractions of human C28I2 cells were extracted with Mem-PER Plus Membrane Protein Extraction Kit (Thermo Fisher Scientific, 89842) and subjected to western blotting analysis. The following primary antibodies were used: Na$_v$1.7 (1:500, Alomone Labs, ASC-008), TNFR2 (1:1,000, ProteinTech, 19272-1-AP), and GAPDH (1:5,000, ProteinTech, 60004-1-Ig).

## ELISA

The levels of HSP70 and midkine in human sera and synovial fluid from healthy individuals and patients with OA, in conditioned medium and cell lysates of human C28I2 cells treated with 25 nM ProTx II or 1 μM PF-04856264 were measured by ELISA according to the manufacture's instructions (Abcam, ab133060, ab193761). Before ELISA analysis, human synovial fluids were digested with hyaluronidase at 1 unit per 100 μl synovial fluid for 1 h at 37 °C. The levels of PRG4 and MMP13 in

supernatant of human OA cartilage explants were measured by ELISA according to the manufacturer's instructions, respectively (R&D systems, DY9829-05; Abcam, ab100605). The levels of HSP70 and midkine in mouse sera and conditioned medium collected from primary chondrocytes were measured by ELISA according to the manufacture's instructions (Abcam, ab133061, ab279416).

## Na⁺ and Ca²⁺ fluorescence imaging

Human OA chondrocytes or C28I2 cells were seeded on 8-well chamber (Thermo Fisher, 154461) and loaded with 5 µM CoroNa Green (Invitrogen, C36676) or Fluo-8 (Abcam, ab112129) in Hanks' Balanced Salt Solution for 45 min at 37 °C in the presence or absence of 25 nM ProTx II or 1 µM PF-04856264 with or without 0.5 µM KB-R7943 (Tocris, 1244). CoroNa Green and Fluo-8 were excited at 488 nm and fluorescence images (525–530 nm) were acquired with 25x water-dipping objective on Zeiss 880 confocal microscope every 2 s during the experiment. Na⁺ and Ca²⁺ transients in chondrocytes were induced by 100 nM ATP in the presence or absence of 25 nM ProTx II or 1 µM PF-04856264. Fluorescence was expressed as the ratio of cytosolic fluorescence and initial intensity ($F/F_0$).

## Measurement of intracellular Ca²⁺ with plate reader

Intracellular Ca²⁺ in human OA chondrocytes or C28I2 cells was measured using Fluo-8 Calcium Flux Assay Kit (Abcam, ab112129) according to the manufacturer's instructions. In brief, cells seeded in black walled 96-well plate (Corning, 3904) were loaded with Fluo-8 in Hanks' Balanced Salt Solution for 30 min at 37 °C and 30 min at room temperature in the presence or absence of 25 nM ProTx II or 1 µM PF-04856264. Ca²⁺ transients in chondrocytes were induced by 100 nM ATP in the presence or absence of 25 nM ProTx II or 1 µM PF-04856264. Fluorescence was measured in a fluorescent plate reader with an excitation wavelength of 490 nm and emission wavelength of 525 nm, and expressed as the ratio of cytosolic fluorescence and initial intensity ($F/F_0$).

## Statistical analysis

All data are presented as mean ± s.d., unless otherwise specified in the figure or table legends. The numbers of mice used per genotype are indicated in figure legends. Comparisons between the two groups were analysed using two-tailed unpaired Student's $t$-test unless stated otherwise in the figure legends. ANOVA with post hoc Bonferroni test was used when comparing multiple groups as described in the figure legends. A value of $P < 0.05$ was considered statistically significant. Statistical analyses were performed using GraphPad Prism 9.

## Reporting summary

Further information on research design is available in the Nature Portfolio Reporting Summary linked to this article.

## Data availability

Full immunoblots are provided as Supplementary Fig. 1. Source data are provided with this paper.

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

**Acknowledgements** We thank Proteomics Laboratory at New York University Grossman School of Medicine for protein-related mass spectrometry analysis. This work was supported partly by NIH research grants R01AR078035, R01AR062207, R01AR061484, R01AR076900 and R01NS070328. Patents have been filed by New York University and Yale University that claim "use of Nav 1.7 inhibitors for preventing joint degeneration and reducing pain in osteoarthritis".

**Author contributions** W.F., S.G.W. and C.-j.L. conceived, designed and directed the study. D.V. recorded and analysed sodium channel currents in human chondrocytes. W.F. performed most of the experiments with the assistance of Y.B., M.Z., G.S., E.K., J.G. and L.Z. (animal behaviour test and histology analysis); A.K. (analysed Na$_v$1.7 mRNA level in human cartilage); W. He (analysed the effects of Na$_v$1.7 blockers on chondrocyte metabolism); S.L. (maintained primary human chondrocyte culture); X.C., M.C., G.H. and Y.L. (establishment of DMM model); W. Hu (analysis of RNA-seq data); G.H., R.Z. and X.Z. (histology analysis); A.H., G.S. and M.Z. (maintenance of mouse colonies, calculation of mouse travel distance in open field test and human sample collection); E.X.G. (intracellular signalling analysis); E.S., P.L. and R.S. (provided human cartilage samples); J.S. and M.A. (provided cDNAs extracted from human normal and arthritic cartilage for RT–qPCR, and sera and synovia fluids from healthy individuals and patients with OA). W.F., D.V., S.G.W. and C.-j.L. wrote the manuscript with inputs from all authors.

**Competing interests** S.G.W. has served as a paid advisor to OliPass, Navega Therapeutics, Sangamo Therapeutics, Chromocell, ThirdRock and Medtronics, and holds stock options in Navega. Compounds and constructs under development by these companies were not used or tested in this study. The remaining authors declare no competing interests.

**Additional information**
**Correspondence and requests for materials** should be addressed to Stephen G. Waxman or Chuan-ju Liu.

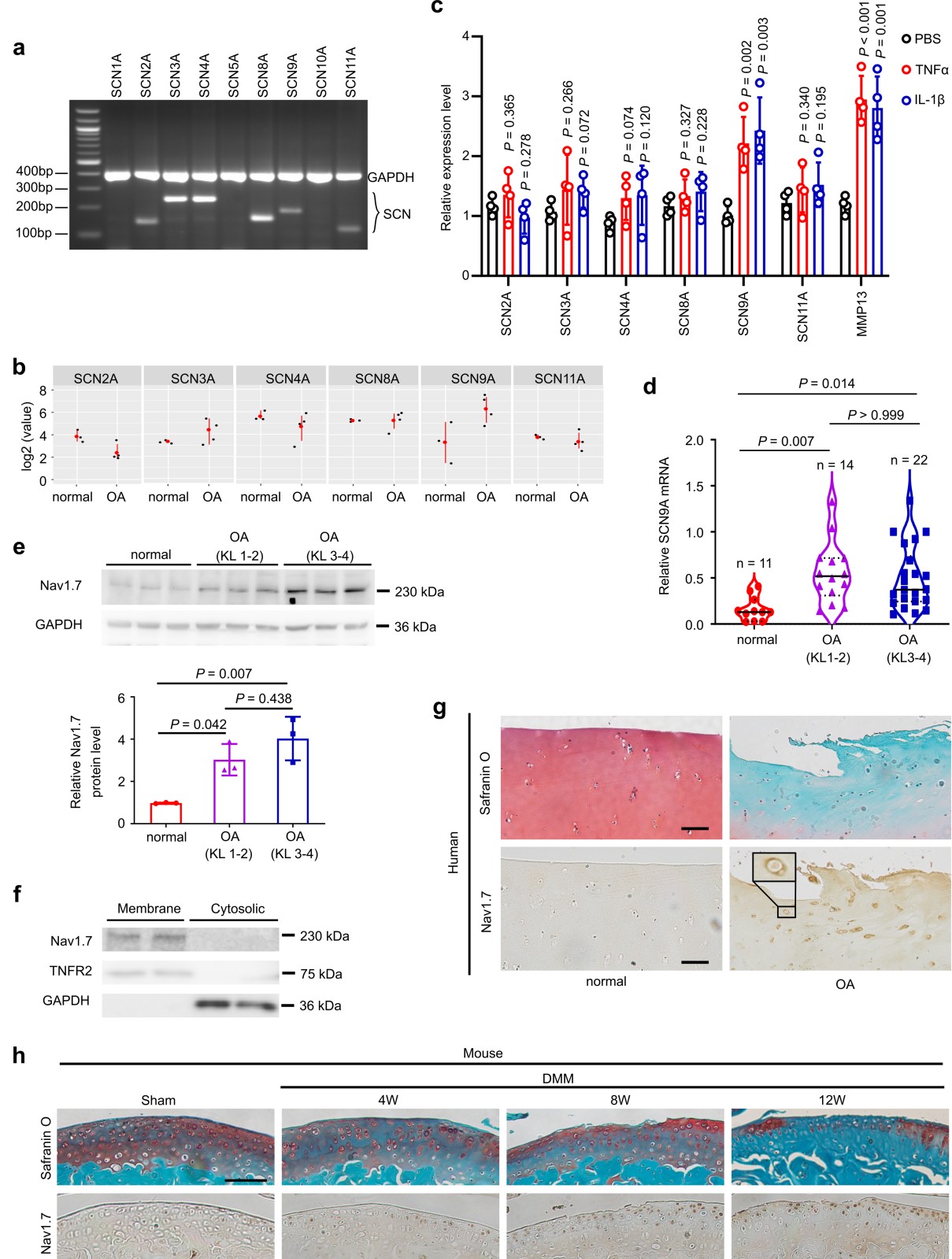

**Extended Data Fig. 1** | See next page for caption.

**Extended Data Fig. 1 | Nav1.7 is expressed in chondrocytes and upregulated in OA cartilage. a**, Relative expression of 9 distinct VGSCs in human C28I2 chondrocytes, assayed by PCR. Expression of *GAPDH* mRNA serves as internal control. **b**, Relative expressions of VGSCs in human OA (n = 4) versus healthy (n = 3) cartilage, in RNA-sequencing data of human cartilage. **c**, The expression levels of chondrocytes expressed VGSCs in human C28I2 cells treated with 10 ng/ml TNFα or IL-1β for 24 h, assayed by qRT-PCR (n = 4 biological replicates). **d**, qPCR analysis of *SCN9A* in human cartilage from healthy individuals (n = 11) and from individuals with early stage (KL grade 1-2, n = 14) or late stage OA (KL grade 3-4, n = 22). **e**, Expression of Nav1.7 in cartilage from healthy individuals and OA patients with KL grade 1-2 or 3-4, assayed by Immunoblotting (n = 3 for each group). **f**, Membrane location of Nav1.7 in human chondrocytes detected by Immunoblotting (n = 3 biological replicates). **g**, Immunostaining of Nav1.7 in cartilage from healthy individuals and from individuals with late stage OA (n = 4 for each group). Scale bar = 100 μm. **h**, Immunostaining of Nav1.7 in cartilage collected from mice subjected to sham or DMM surgery (n = 4 mice per group). Scale bar = 100 μm. Data are means ± SD, *P* values are calculated by two-tailed unpaired Student's t-test in **c** and one way ANOVA with Bonferroni post-hoc test in **d** and **e**. DMM, destabilization of medial meniscus; OA, osteoarthritis.

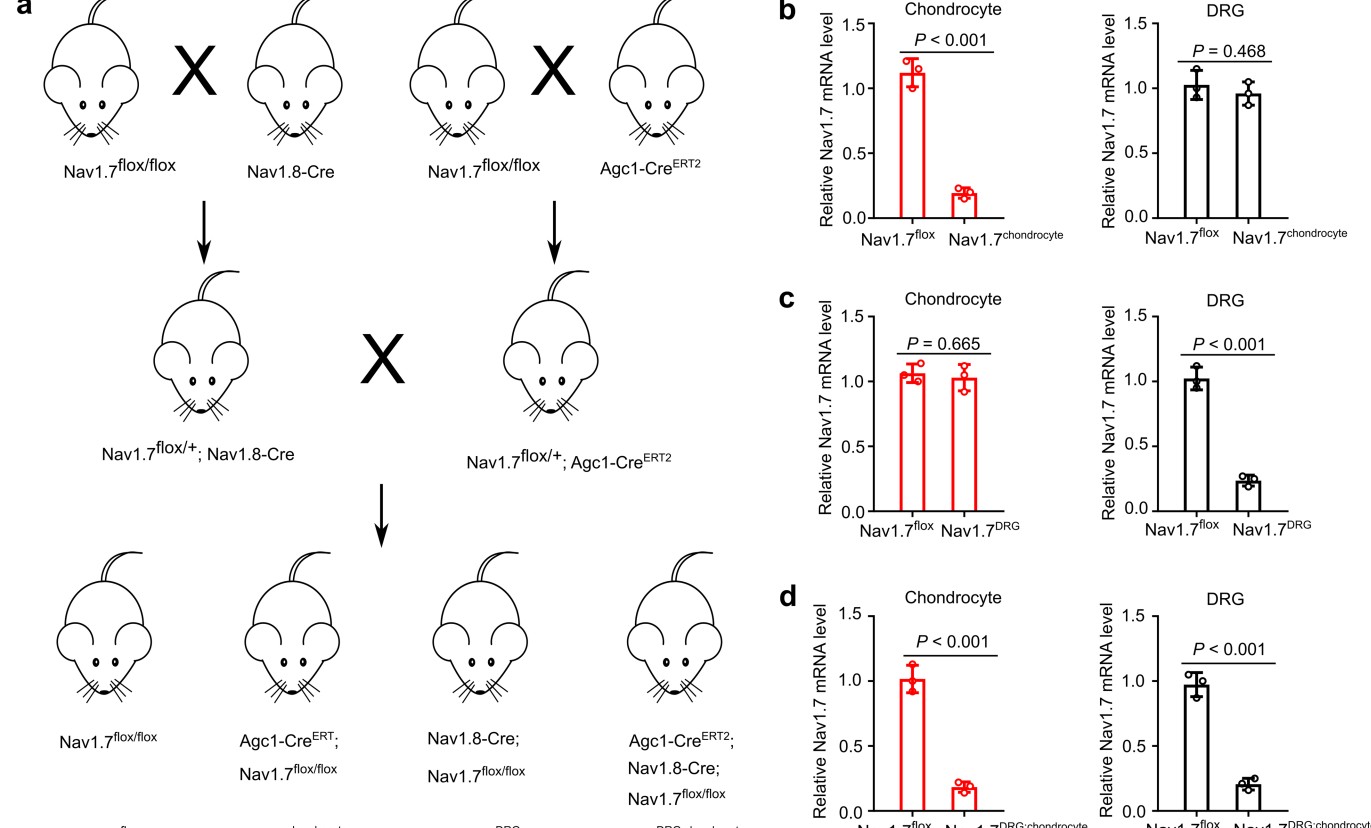

**Extended Data Fig. 2 | Generation of conditional Nav1.7 knockout mouse.**
**a**, Schematic of mouse breeding strategy to generate conditional knockout mice lacking Nav1.7 in chondrocyte (Nav1.7chondrocyte), DRG neurons (Nav1.7DRG) or both chondrocyte and DRG neurons (Nav1.7DRG;chondrocyte). **b-d**, Relative *Nav1.7* mRNA level in chondrocyte and DRG neurons from Nav1.7chondrocyte (**b**), Nav1.7DRG (**c**), and Nav1.7DRG;chondrocyte (**d**) mice (n = 3 biological replicates). 10-week-old mice Agc1-CreERT2;Nav1.7flox/flox and Agc1-CreERT2;Nav1.8-Cre;Nav1.7flox/flox were intraperitoneally injected with tamoxifen at a dose of 150 μg per gram of body weight, administered daily for 5 consecutive days. Basal levels of Nav1.7 in DRG neurons and chondrocytes from Nav1.7flox mice were set as 1, respectively. Data are means ± SD, *P* values are calculated by two-tailed unpaired Student's t-test.

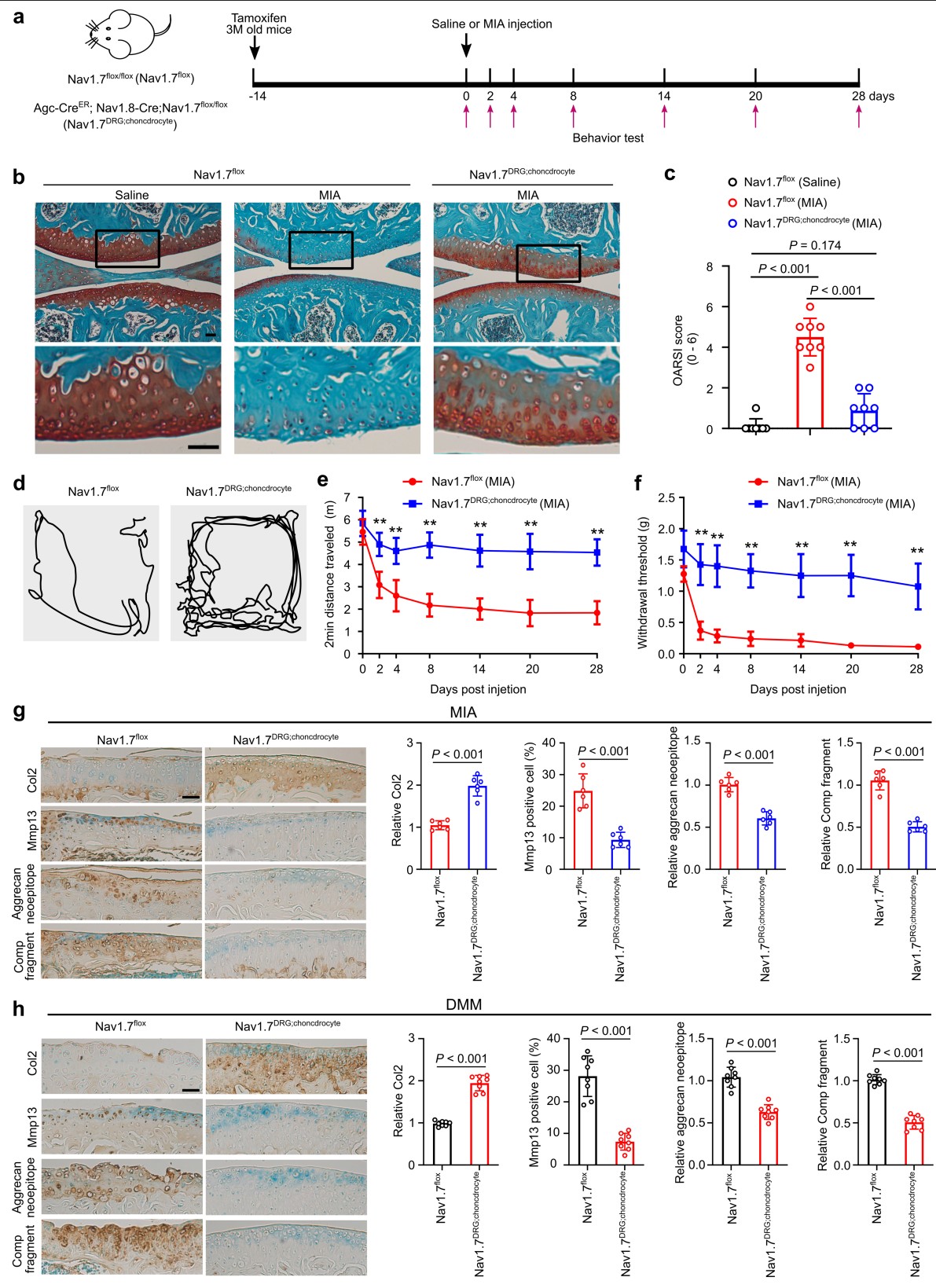

**Extended Data Fig. 3** | See next page for caption.

**Extended Data Fig. 3 | Genetic ablation of Nav1.7 in both chondrocyte and DRG neuron ameliorates OA progression and alleviates pain in chemically induced OA model. a**, Experimental scheme showing establishment of MIA model in mice harboring tamoxifen inducible Nav1.7 deletion in chondrocyte and DRG neuron. **b**, Representative Safranin O/fast green stained knee joint sections of Nav1.7$^{flox}$ and Nav1.7$^{DRG;chondrocyte}$ male after saline or MIA injection at day 28 (n = 8 mice for each group). Scale bar = 50 μm. **c**, Quantification of OARSI score shown in **b. d**, Representative traces of open field testing at day 28 post MIA injection in Nav1.7$^{flox}$ and Nav1.7$^{DRG;chondrocyte}$ mice. **e, f**, Quantitation of 2 min travel distance (**e**) and von Frey testing (**f**) in Nav1.7$^{flox}$ and Nav1.7$^{DRG}$ mice at the indicated time-points post MIA injection (n = 8 mice per group). **g**, Immunohistochemical staining and corresponding quantification for COL2, Mmp13, Aggrecan neoepitope, and Comp fragment in knee joint sections of Nav1.7$^{flox}$ and Nav1.7$^{DRG;chondrocyte}$ mice at day 28 post MIA injection (n = 6). Scale bar = 50 μm. **h**, (**related to** Fig. 2a–h) Immunohistochemical staining and corresponding quantification of COL2, Mmp13, aggrecan neoepitope, and Comp fragment in knee joint sections of Nav1.7$^{flox}$ and Nav1.7$^{DRG; chondrocyte}$ mice at 12 weeks post DMM surgery (n = 8). Scale bar = 50 μm. Data are means ± SD, $P$ values are calculated by one way ANOVA with Bonferroni post-hoc test (**c**) and two-tailed unpaired Student's t-test (**g, h**). Data are means ± 95% confidence interval (CI), $P$ values are calculated by two-tailed multiple unpaired Student's t-test with Welch correction (**e, f**). * $P < 0.05$; ** $P < 0.01$.

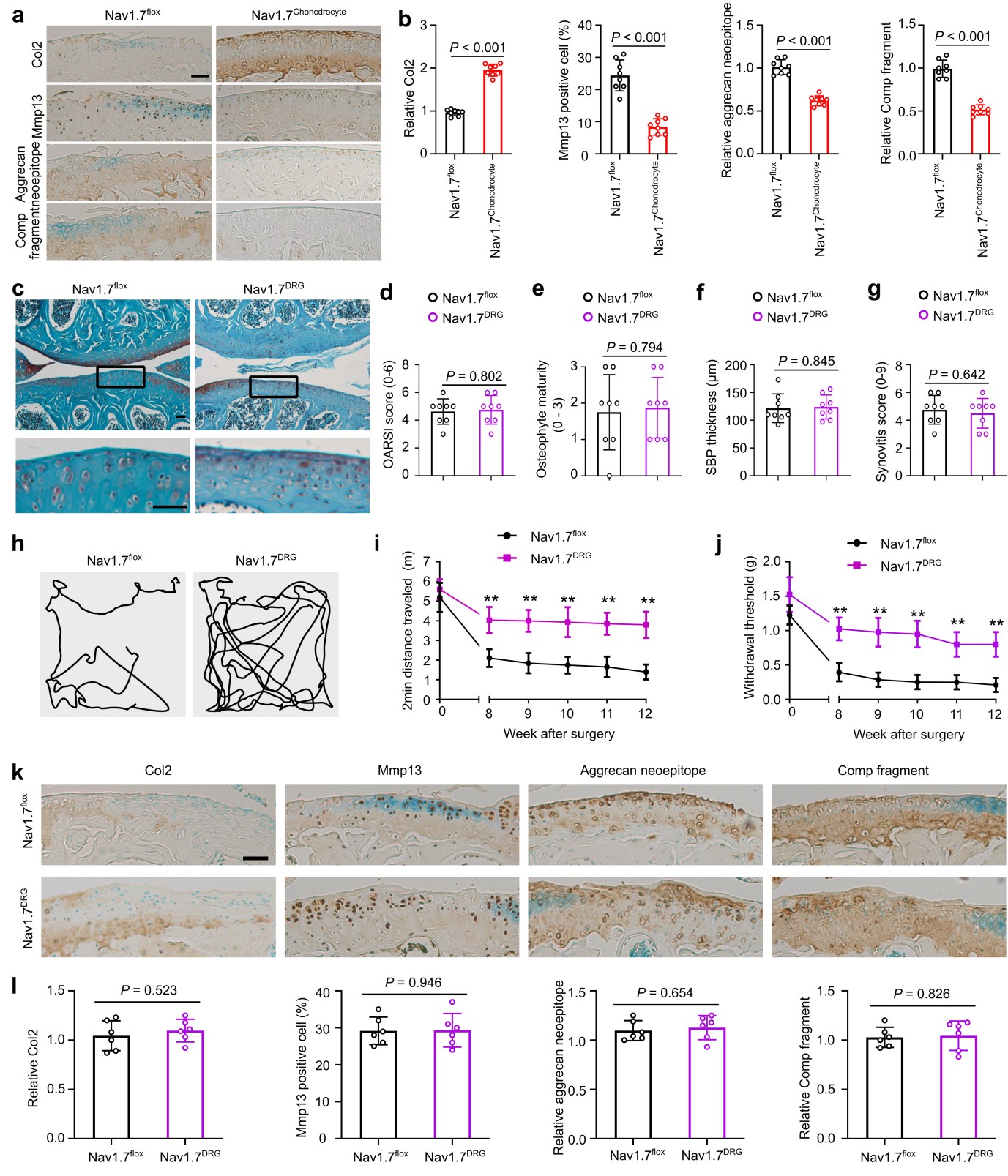

**Extended Data Fig. 4** | See next page for caption.

**Extended Data Fig. 4 | Genetic ablation of Nav1.7 in DRG neuron contributes to reduce OA associated pain without affecting OA pathological progression in surgically induced OA model. a**, **b**, (**related to** Fig. 2i–p) Immunohistochemical staining (**a**) and corresponding quantification (**b**) of Col2, Mmp13, Aggrecan neoepitope, and Comp fragment in knee joint sections of Nav1.7$^{flox}$ and Nav1.7$^{Chondrocyte}$ mice at 12 weeks post DMM surgery (n = 8). Scale bar = 50 μm in **a**. **c**, Representative Safranin O/fast green stained sections of knee joints from DMM operated Nav1.7$^{flox}$ and Nav1.7$^{DRG}$ mice (n = 8 mice per group). Scale bar = 50 μm. **d**-**g**, Quantitation of OARSI score (**d**), osteophyte development (**e**), SBP thickness (**f**), and synovitis score (**g**) in Nav1.7$^{flox}$ and Nav1.7$^{DRG}$ mice subjected to DMM surgery at week 12 (n = 8 mice per group). **h**, Representative traces of open field testing at 12 weeks post DMM surgery in Nav1.7$^{flox}$ and Nav1.7$^{DRG}$ mice. **i**, **j**, Quantitation of 2 min travel distance (**i**) and von Frey testing (**j**) in DMM operated Nav1.7$^{flox}$ and Nav1.7$^{DRG}$ mice at the indicated time-points after surgery (n = 8 mice per group). **k**, **l**, Immunohistochemical staining (**k**) and corresponding quantification (**l**) for Col2, Mmp13, Aggrecan neoepitope, and Comp fragment in knee joint sections of Nav1.7$^{flox}$ and Nav1.7$^{DRG}$ mice at 12 weeks post DMM surgery (n = 6). Scale bar = 50 μm in **k**. Data are means ± SD, *P* values are calculated by two-tailed unpaired Student's t-test in b, d-g and l; data are means ± 95% confidence interval (CI), *P* values are calculated by two-tailed multiple unpaired Student's t-test with Welch correction in i and j (** *P* < 0.01).

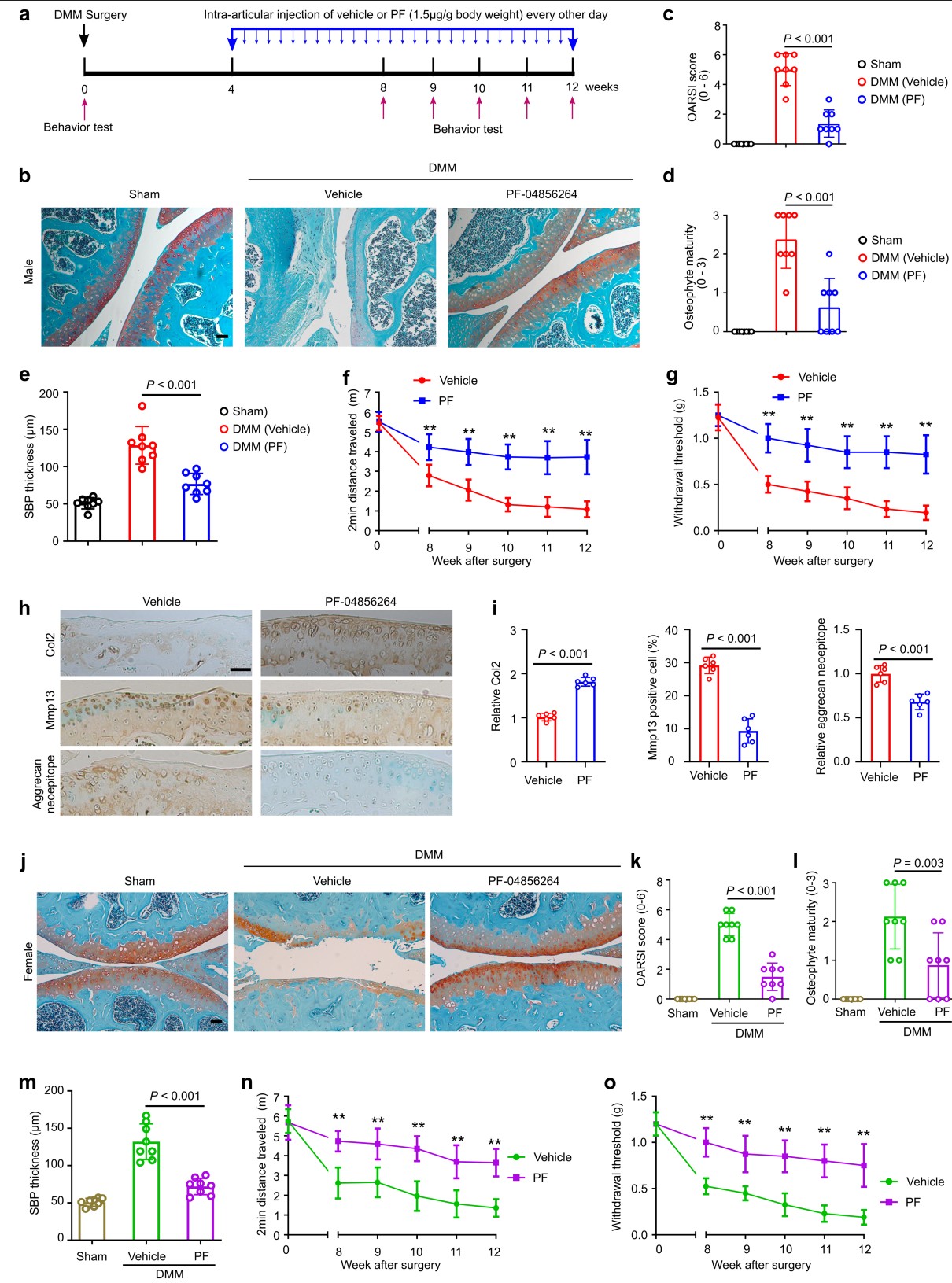

**Extended Data Fig. 5** | See next page for caption.

**Extended Data Fig. 5 | Pharmacological blockade of Nav1.7 through locally is therapeutic against OA and OA-related pain in surgically induced mouse models. a**, Schematic of the experimental design to determine the effects of intra-articular administrated PF-04856264 (the selective Nav1.7 inhibitor) on OA progression and pain-related behaviors in the surgically induced DMM mouse model. **b**, Representative Safranin O/fast green stained knee joint sections of DMM operated WT male mice treated with or without PF-04856264 for 8 weeks (n = 8 mice for each group). Scale bar = 50 μm. **c-e**, Quantitation of OARSI score (**c**), osteophyte development (**d**) and SBP thickness (**e**) in DMM operated WT male mice treated with or without PF-04856264 for 8 weeks (n = 8 mice per group). **f, g**, Quantitation of 2 min travel distance (**f**) and von Frey testing (**g**) in DMM-operated WT male mice treated with or without PF-04856264 at the indicated time-points after surgery (n = 8 mice per group). **h, i**, Immunohistochemical staining (**h**) and corresponding quantification (**i**) of Col2, Mmp13, and Aggrecan neoepitope in knee joint sections of DMM operated WT male mice treated with or without PF-04856264 for 8 weeks (n = 6). Scale bar = 50 μm in **h. j**, Representative Safranin O/fast green stained knee joint sections of DMM operated WT female mice treated with or without PF-04856264 for 8 weeks (n = 8 mice for each group). Scale bar = 50 μm. **k-m**, Quantitation of OARSI score (**k**), osteophyte development (**l**) and SBP thickness (**m**) in DMM operated WT female mice treated with or without PF-04856264 for 8 weeks (n = 8 mice per group). **n, o**, Quantitation of 2 min travel distance (**n**) and von Frey testing (**o**) in DMM-operated WT female mice treated with or without PF-04856264 at the indicated time-points after surgery (n = 8 mice per group). Data are means ± SD, *P* values are calculated by one way ANOVA with Bonferroni post-hoc test (**c**, **d**, **e**, **k**, **l** and **m**) and two-tailed unpaired Student's t-test (**i**). Data are means ± 95% confidence interval (CI), *P* values are calculated by two-tailed multiple unpaired Student's t-test with Welch correction in f, g, n and o (**$P$ < 0.01).

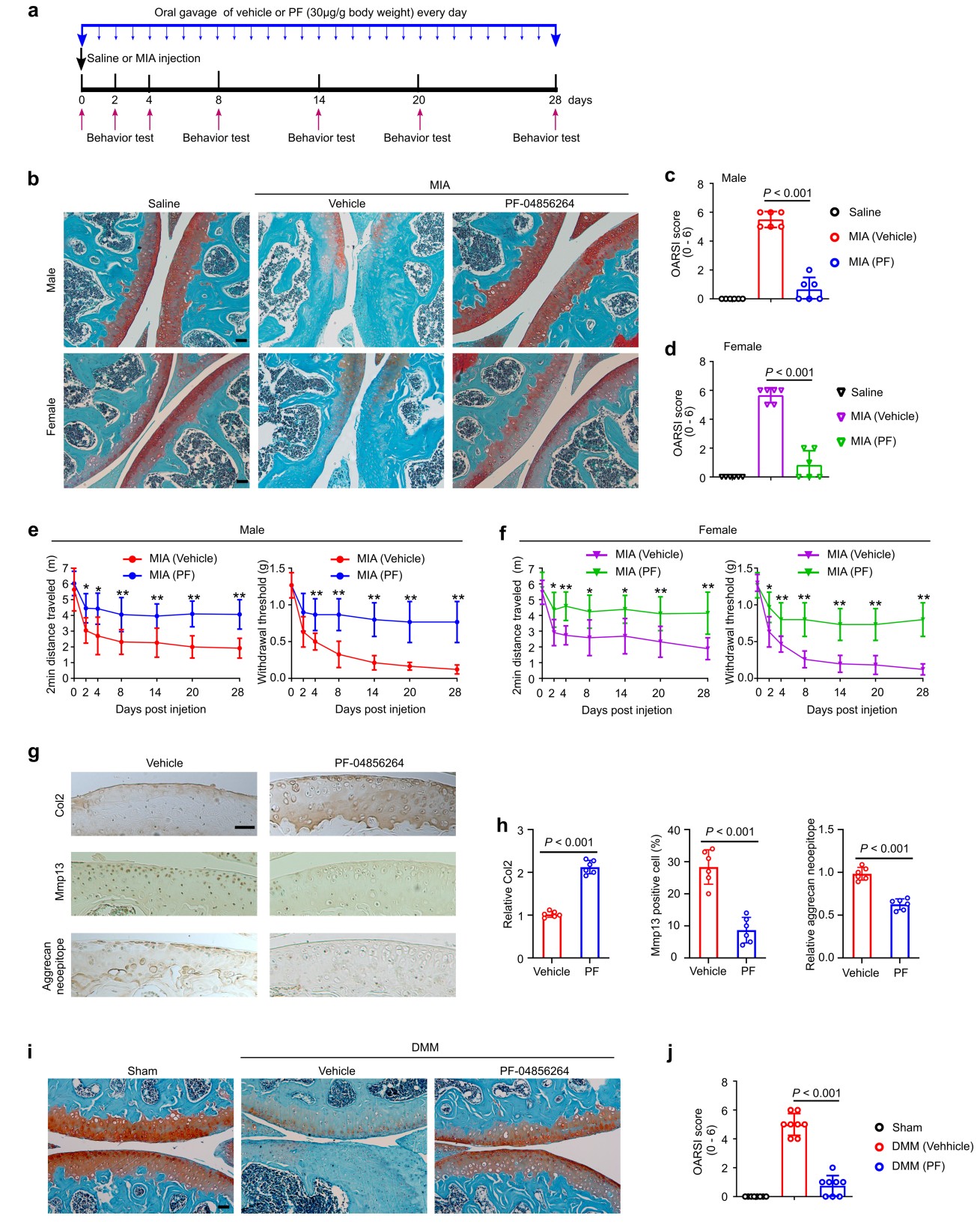

**Extended Data Fig. 6** | See next page for caption.

**Extended Data Fig. 6 | Oral delivery of PF-04756264 attenuates OA progression and alleviates pain in both chemically and surgically induced OA models. a**, A schematic representation highlights the experimental outline to determine the effects of PF-04856264 through oral delivery on OA progression and behavior changes in WT male and female mice with MIA model. **b**, Representative Safranin O/fast green stained knee joint sections of WT male and female mice treated with or without PF-04856264 after saline or MIA injection at day 28 (n = 6 mice for each group). Scale bar = 50 μm. **c**, **d**, Quantification of OARSI score in male (**c**) and female (**d**) mice as shown in **b**. **e**, **f**, Quantitation of 2 min travel distance and von Frey testing in WT male (**e**) and female (**f**) mice treated with or without PF-04856264 at the indicated time-points post MIA injection (n = 6 mice per group). **g**, **h**, Immunohistochemical staining (**g**) and corresponding quantification (**h**) for Col2, Mmp13, and Aggrecan neoepitope in knee joint sections of WT male mice treated with or without PF-04856264 at day 28 post MIA injection (n = 6). Scale bar = 50 μm in **g**. **i**, Representative Safranin O/fast green stained knee joint sections of DMM operated WT male mice orally treated without or with PF-04856264 for 8 weeks (n = 8 mice for each group). Scale bar = 50 μm. **j**, Quantitation of OARSI score for **i**. Data are means ± SD, $P$ values are calculated by one way ANOVA with Bonferroni post-hoc test (**c**, **d** and **j**) and two-tailed unpaired Student's t-test (**h**). Data are means ± 95% confidence interval (CI), $P$ values are calculated by two-tailed multiple unpaired Student's t-test with Welch correction in e and f (*$P < 0.05$; **$P < 0.01$).

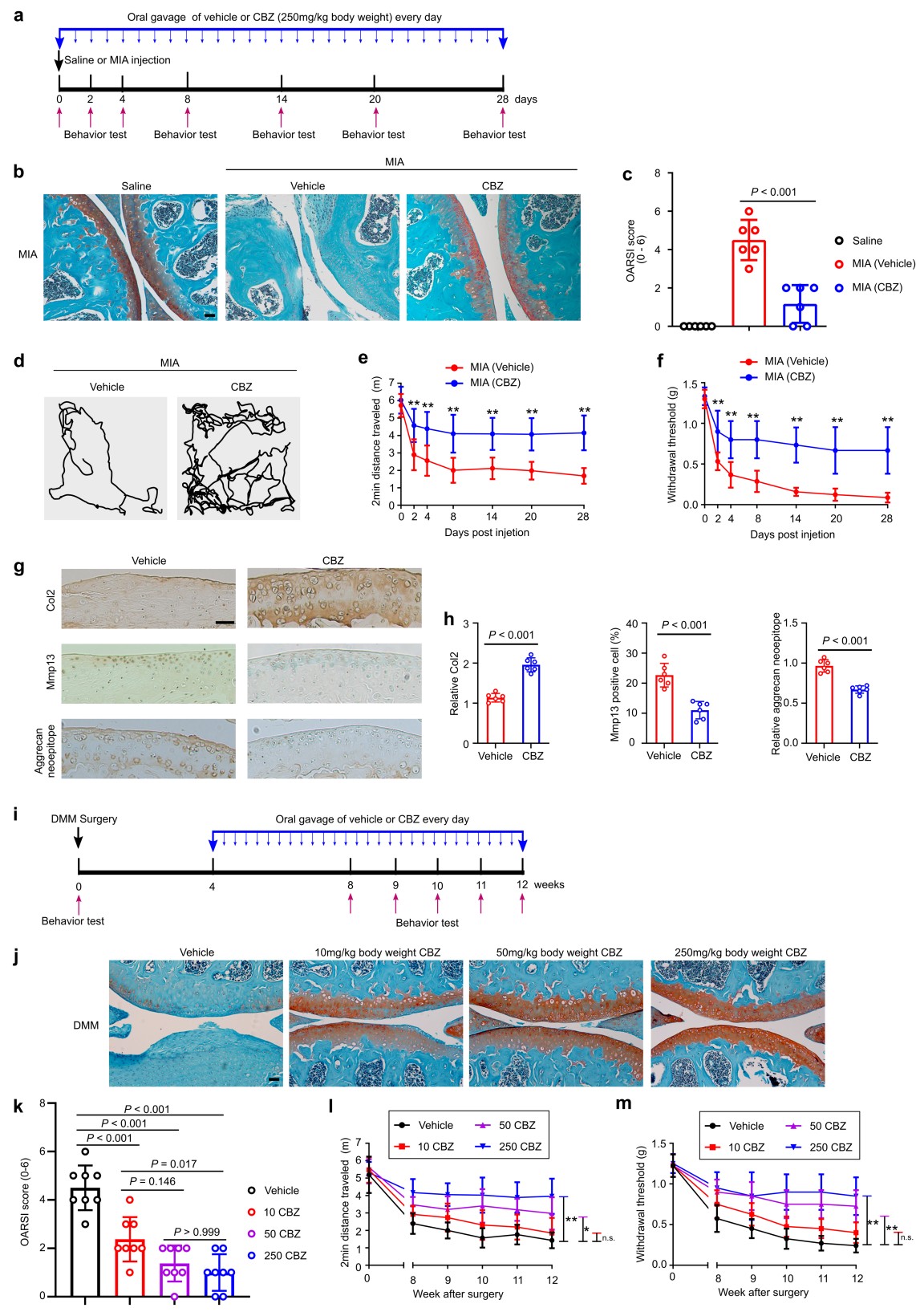

**Extended Data Fig. 7 |** See next page for caption.

**Extended Data Fig. 7 | Carbamazepine (CBZ), a clinically used sodium chancel blocker, attenuates OA progression and pain in chemically and surgically induced OA models. a**, Schematic of timeline to analyze the effects of systemic oral delivery carbamazepine (CBZ) on OA progression and pain in MIA model. **b**, Representative Safranin O/fast green stained knee joint sections of WT male mice treated with or without CBZ after saline or MIA injection at day 28 (n = 6 mice for each group). Scale bar = 50 μm. **c**, Quantification of OARSI score as shown in **b**. **d**, Representative traces of open field testing at day 28 post MIA injection in WT male mice with indicated treatment at day 28 post MIA injection. **e**, **f**, Quantitation of 2 min travel distance (**e**) and von Frey testing (**f**) in WT male mice at the indicated time-points post MIA injection (n = 6 mice per group). **g**, **h**, Immunohistochemical staining (**g**) and corresponding quantification (**h**) for Col2, Mmp13, and Aggrecan neoepitope in knee joint sections of WT mice with indicated treatment at day 28 post MIA injection (n = 6). Scale bar = 50 μm in **g**. **i**, Schematic of timeline to analyze the effects of systemic oral delivery of various dosages of CBZ on OA progression and pain in surgically induced DMM model. **j**, Representative Safranin O/fast green stained knee joint sections of DMM operated WT male mice treated without or with 10 mg/kg body weight, 50 mg/kg body weight or 250 mg/kg body weight CBZ for 8 weeks (n = 8 mice for each group). Scale bar = 50 μm. **k**, Quantitation of OARSI score for **j**. **l**, **m**, Quantitation of 2 min travel distance (**l**) and von Frey testing (**m**) in DMM-operated WT male mice treated without or with various dosage of CBZ, as indicated, at specified time-points after surgery (n = 8 mice per group). Data are means ± SD, *P* values are calculated by one way ANOVA with Bonferroni post-hoc test (**c**, **k**) and two-tailed unpaired Student's t-test (**h**). Data are means ± 95% confidence interval (CI), *P* values are calculated by two-tailed multiple unpaired Student's t-test with Welch correction in (**e** and **f**) and two way ANOVA with Bonferroni post-hoc test (**l**, **m**). *$P$ < 0.05; **$P$ < 0.01.

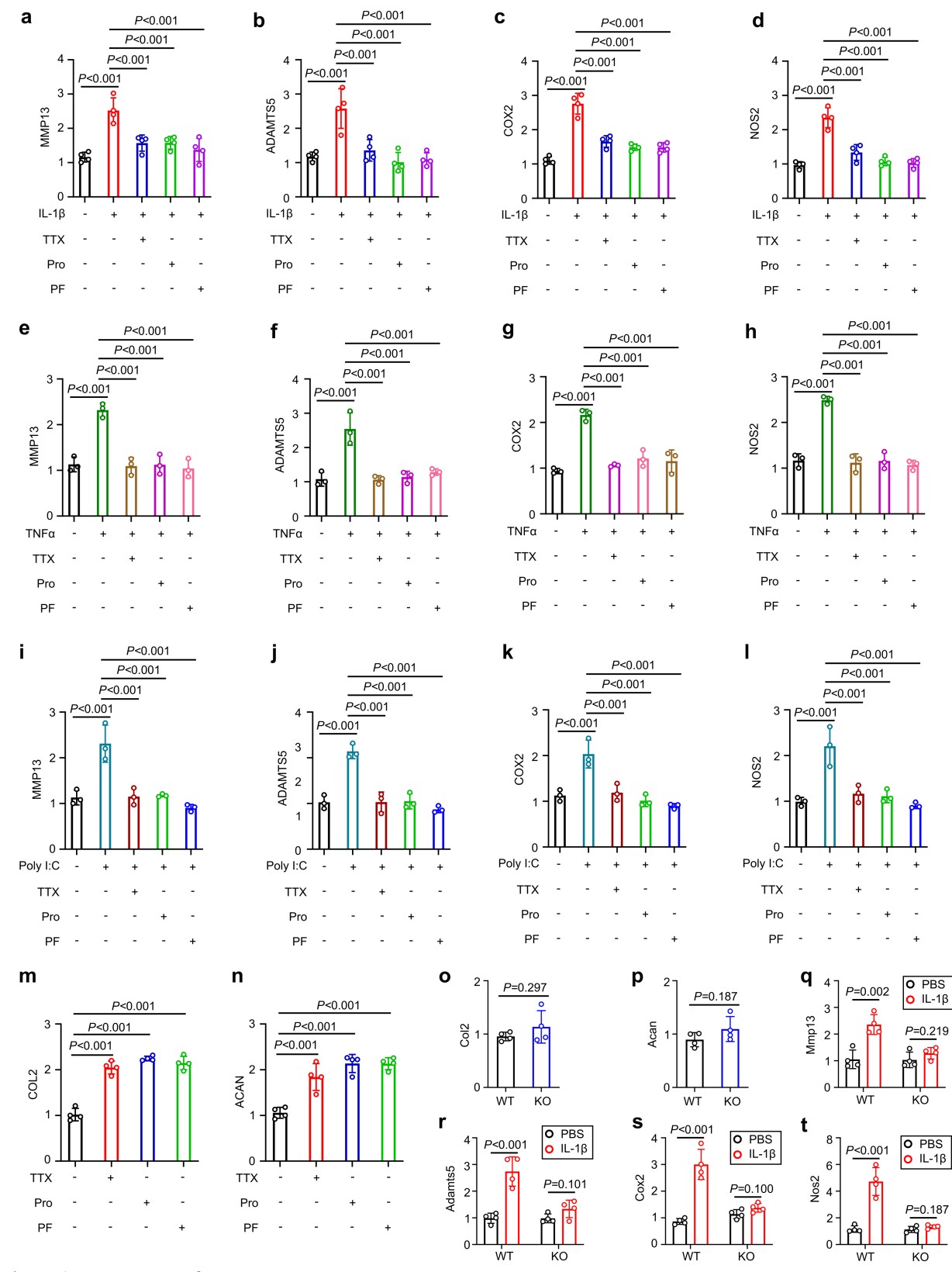

**Extended Data Fig. 8** | See next page for caption.

**Extended Data Fig. 8 | Pharmacological blockade or deletion of Nav1.7 enhances anabolic and inhibits catabolic metabolism in chondrocytes.** **a**-**d**, mRNA levels of catabolism markers *MMP13* (**a**), *ADAMTS5* (**b**), *COX2* (**c**) and *NOS2* (**d**) in human C28I2 chondrocytes treated with or without 10 ng/ml IL-1β in the absence or presence of 1 μM TTX, 25 nM ProTx II (Pro) or 1 μM PF-04856264 (PF) for 24 h, assayed by qRT-PCR (n = 4 biological replicates). **e**-**h**, mRNA levels of catabolism markers *MMP13* (**e**), *ADAMTS5* (**f**), *COX2* (**g**) and *NOS2* (**h**) in human C28I2 chondrocytes treated with or without 10 ng/ml TNFα in the absence or presence of 1 μM TTX, 25 nM ProTx II (Pro) or 1 μM PF-04856264 (PF) for 24 h, assayed by qRT-PCR (n = 3 biological replicates). **i**-**l**, mRNA levels of catabolism markers *MMP13* (**i**), *ADAMTS5* (**j**), *COX2* (**k**) and *NOS2* (**l**) in human C28I2 chondrocytes treated with or without 500 ng/ml poly(I:C) in the absence or presence of 1 μM TTX, 25 nM ProTx II (Pro) or 1 μM PF-04856264 (PF) for 24 h, assayed by qRT-PCR (n = 3 biological replicates). **m**, **n**, mRNA levels of anabolic markers *COL2* (**m**) and *ACAN* (**n**) in human C28I2 chondrocytes treated with 1 μM TTX, 25 nM ProTx II (Pro) or 1 μM PF-04856264 (PF) for 24 h, assayed by qRT-PCR (n = 4 biological replicates). **o**, **p**, mRNA levels of *Col2* (**o**) and *Acan* (**p**) in chondrocytes isolated from Nav1.7[flox] and Nav1.7[chondrocyte] mice at P6 (n = 4 biological replicates). **q**-**t**, mRNA levels of *Mmp13* (**q**), *Adamts5* (**r**), *Cox2* (**s**) and *Nos2* (**t**) in chondrocytes isolated from Nav1.7[flox] and Nav1.7[chondrocyte] mice at P6 which are treated with or without IL-1β for 24 h (n = 4 biological replicates). Data are mean ± SD, *P* values are calculated by one way ANOVA with Bonferroni post-hoc test (**a**-**n**) and two-tailed unpaired Student's t-test (**o**-**t**).

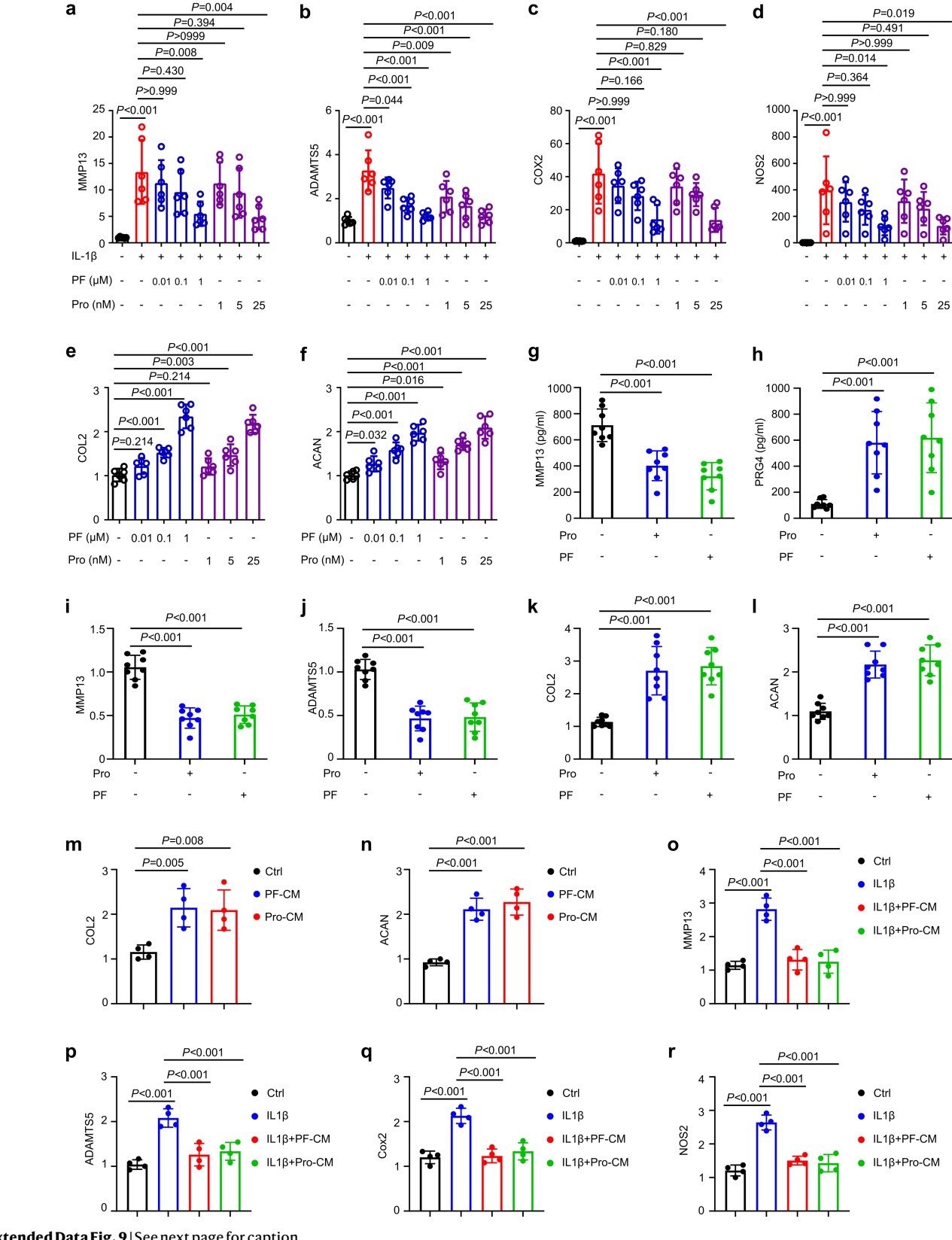

**Extended Data Fig. 9** | See next page for caption.

**Extended Data Fig. 9 | Blocking Nav1.7 pharmacologically inhibit catabolism and enhance anabolism through regulating the chondrocyte secretion.** **a-d**, mRNA levels of *MMP13* (**a**), *ADAMTS5* (**b**), *COX2* (**c**) and *NOS2* (**d**) in primary human chondrocytes isolated from patients with last stage OA which are treated with or without 10 ng/ml IL-1β along with a serial doses of ProTx II (Pro) or PF-04856264 (PF) for 24 h, assayed by qRT-PCR (n = 6 donors). **e, f**, mRNA levels of *COL2* (**e**) and *ACAN* (**f**) in primary human chondrocytes isolated from patients with late stage OA which are treated with a serial doses of ProTX II (Pro) or PF-04856264 (PF) for 24 h, assayed by qRT-PCR (n = 6 donors). **g, h**, Proteins levels of MMP13 (**g**) and PRG4 (**h**) in the supernatants of full-thickness human OA cartilage explants, as determined by ELISA (n = 8 donors). Cartilage explant was cultured with 10 ng/ml IL-1β in the absence or presence of 25 nM ProTx II or 1 μM PF-04856264 for 5 days. **i-l**, mRNA levels of *MMP13* (**i**), *ADAMTS5* (**j**), *COL2* (**k**), and *ACAN* (**l**) in the human OA cartilage explants cultured with 10 ng/ml IL-1β in the absence or presence of 25 nM ProTx II or 1 μM PF-04856264 for 5 days, assayed by qRT-PCR (n = 8 donors). **m, n**, mRNA levels of *COL2* (**m**) and *ACAN* (**n**) in human C28I2 cells treated with conditioned medium collected from 25 nM Pro or 1 μM PF treated C28I2 cells (n = 4 biological replicates). **o-r**, mRNA levels of *MMP13* (**o**), *ADAMTS5* (**p**), *COX2* (**q**) and *NOS2* (**r**) in human C28I2 cells treated with conditioned medium collected from 25 nM Pro or 1 μM PF treated C28I2 cells as well as 10 ng/ml IL-1β for 24 h (n = 4 biological replicates). Data are mean ± SD, *P* values are calculated by one way ANOVA with Bonferroni post-hoc test.

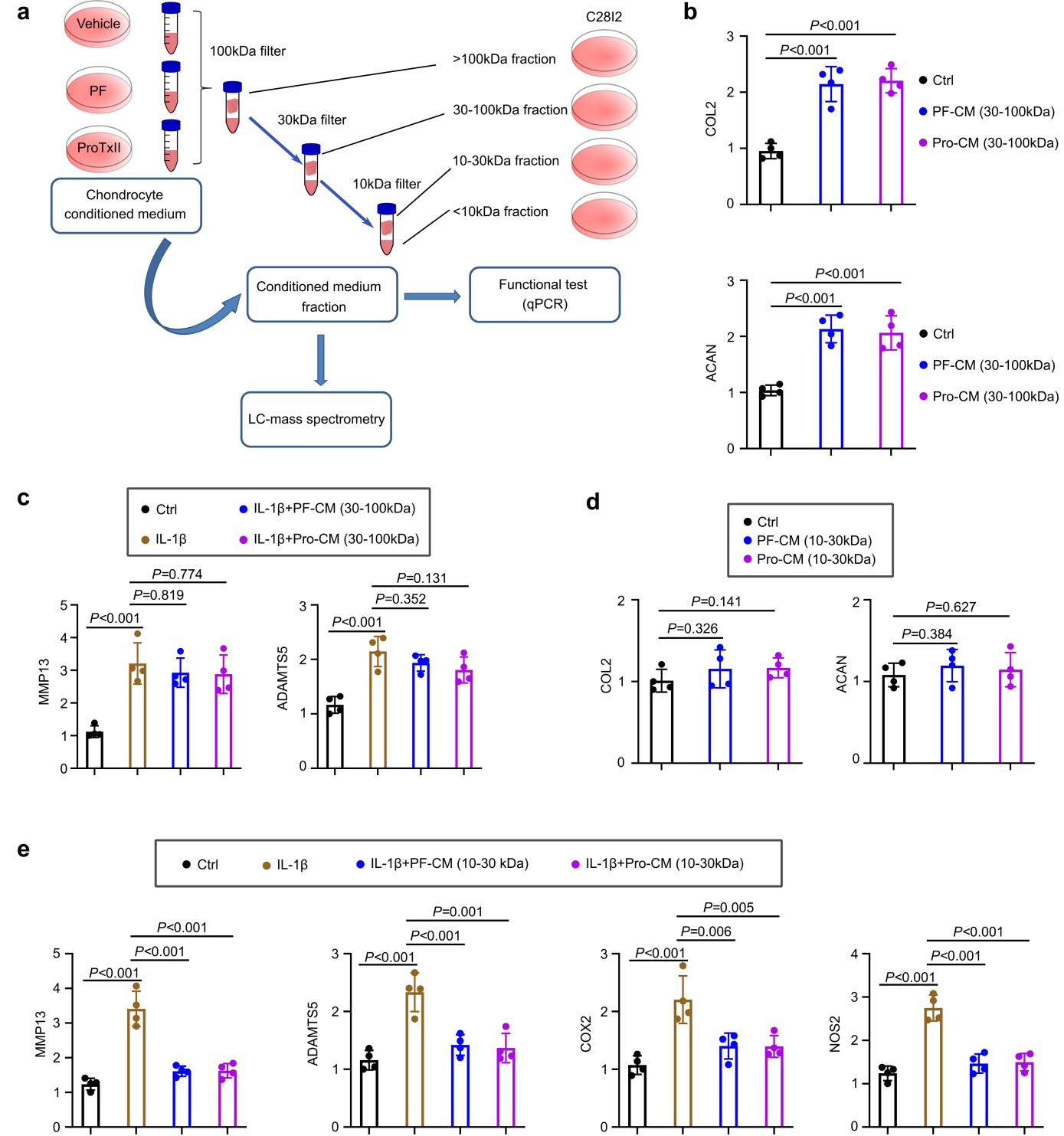

**Extended Data Fig. 10 | Effects of conditioned medium on human chondrocyte anabolism and catabolism. a**, Schematic of the experimental strategy of conditioned medium production, fraction and characterization. Human C28I2 chondrocytes are expanded in DMEM supplemented with FBS until they are 90% confluent. The growth medium is them exchanged with medium supplemented with ITS and 25 nM ProTX II (Pro) or 1 µM PF-04856264 (PF) for 2 days. The medium is then collected and separated into 4 fractions based on sized exclusion: <10 kDa, 10–30 kDa, 30–100 kDa, and >100 kDa using the centrifugal filter device. **b**, mRNA levels of *COL2 and ACAN* in human C28I2 cells treated with 30–100 kDa fraction of the conditioned medium for 24hrs (n = 4 biological replicates). **c**, mRNA levels of *MMP13* and *ADAMTS5* in human C28I2 cells treated with 30–100 kDa fraction of the conditioned medium in the absence or presence of 10 ng/ml IL-1β for 24hrs (n = 4 biological replicates). **d**, mRNA levels of *COL2* and *ACAN* in human C28I2 cells treated with 10–30 kDa fraction of the conditioned medium for 24hrs (n = 4 biological replicates). **e**, mRNA levels of *MMP13, ADAMTS5, COX2* and *NOS2* in human C28I2 cells treated with 10–30 kDa fraction of the conditioned medium in the absence or presence of 10 ng/ml IL-1β for 24hrs (n = 4 biological replicates). Data are means ± SD, *P* values are calculated by one way ANOVA with Bonferroni post-hoc test (**b-e**).

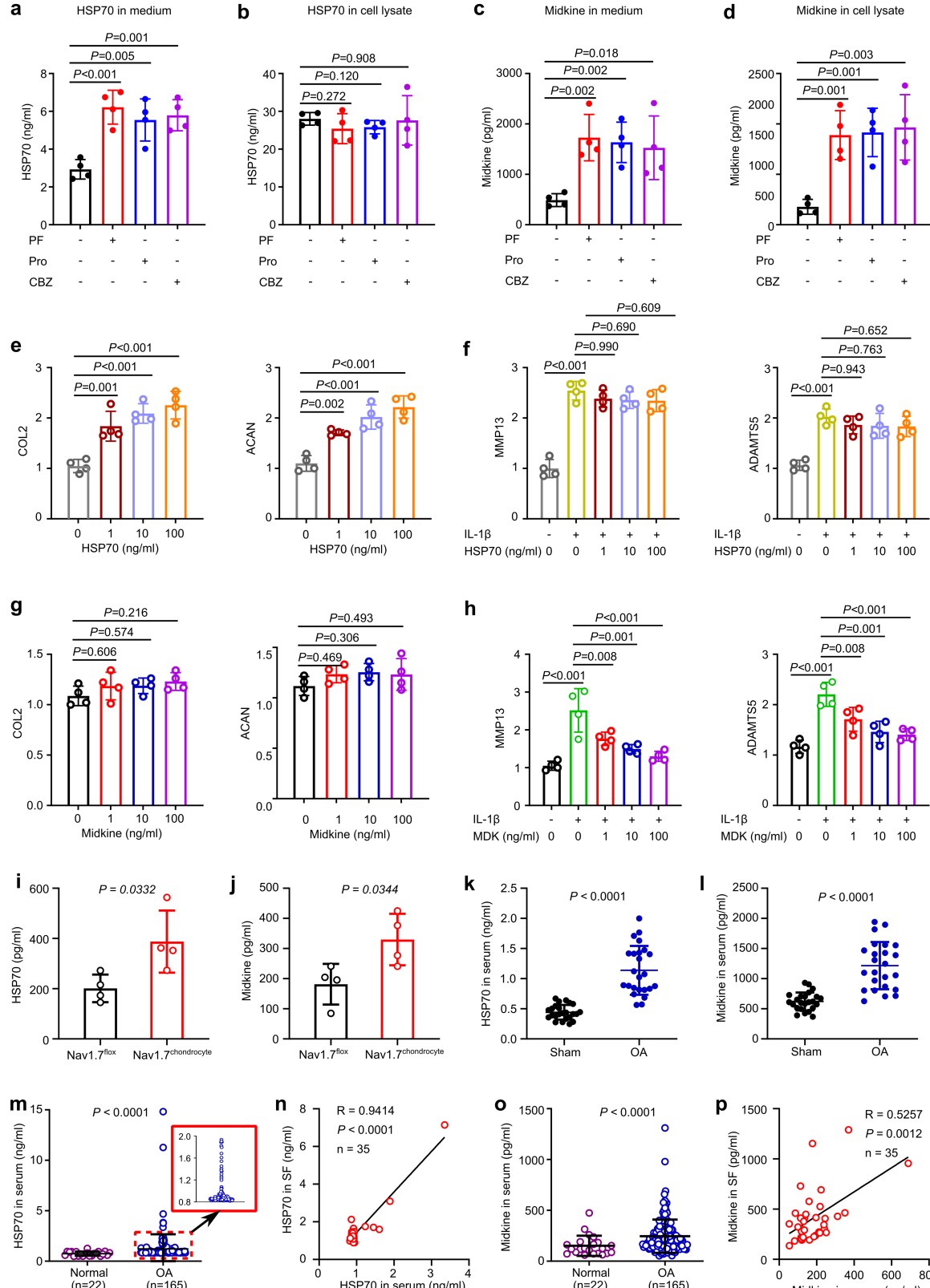

**Extended Data Fig. 11** | See next page for caption.

**Extended Data Fig. 11 | HSP70 enhances anabolism and midkine inhibits IL-1β induced catabolism in human chondrocytes. a**, **b**, ELISA quantification of HSP70 in conditioned medium (**a**) and cell lysate (**b**) of human C28I2 cells treated with 25 nM ProTx II (Pro), 1 μM PF-04856264 (PF), or 10 μM CBZ for 48 h. **c**, **d**, ELISA quantification of Midkine in conditioned medium (**c**) and cell lysate (**d**) of human C28I2 cells treated with 25 nM Pro or 1 μM PF for 48 h. **e**, mRNA levels of *COL2* and *ACAN* in human C28I2 cells treated with serial doses of HSP70 for 24 h. **f**, mRNA levels of *MMP13* and *ADAMTS5* in human C28I2 cells treated with serial doses of HSP70 and 10 ng/ml IL-1β for 24 h. **g**, mRNA levels of *COL2* and *ACAN* in human C28I2 cells treated with serial doses of midkine for 24 h. **h**, mRNA levels of *MMP13* and *ADAMTS5* in human C28I2 cells treated with serial doses of midkine and 10 ng/ml IL-1β for 24 h. n = 4 biological replicates;

Data are mean ± SD, *P* values are calculated by. **i**, **j**, ELISA quantification of HSP70 (**i**) and midkine (**j**) in conditioned medium of primary chondrocytes isolated from Nav1.7$^{flox}$ and Nav1.7$^{chondrocye}$ mice at 12 weeks post DMM surgery (n = 4 biological replicates). **k**, **l**, Levels of HSP70 (**k**) and midkine (**l**) in mouse sera collected from sham surgery control and from DMM surgery WT mice at 12 weeks after surgery (n = 24), assayed by ELISA. **m**, **o**, Levels of HSP70 (**m**) and midkine (**o**) in human sera collected from healthy individuals (n = 22) and from patients with OA (n = 165), assayed by ELISA. **n**, **p**, Correlation analysis (Pearson R and two-tailed P value) of HSP70 (**n**) and midkine (**p**) between matched sera and synovium fluids isolated from OA patients (n = 35). Data are means ± SD, *P* values are calculated by one way ANOVA with Bonferroni post-hoc test (**a**-**h**) and two-tailed unpaired Student's t-test (**i**-**l**, **m**, **o**).

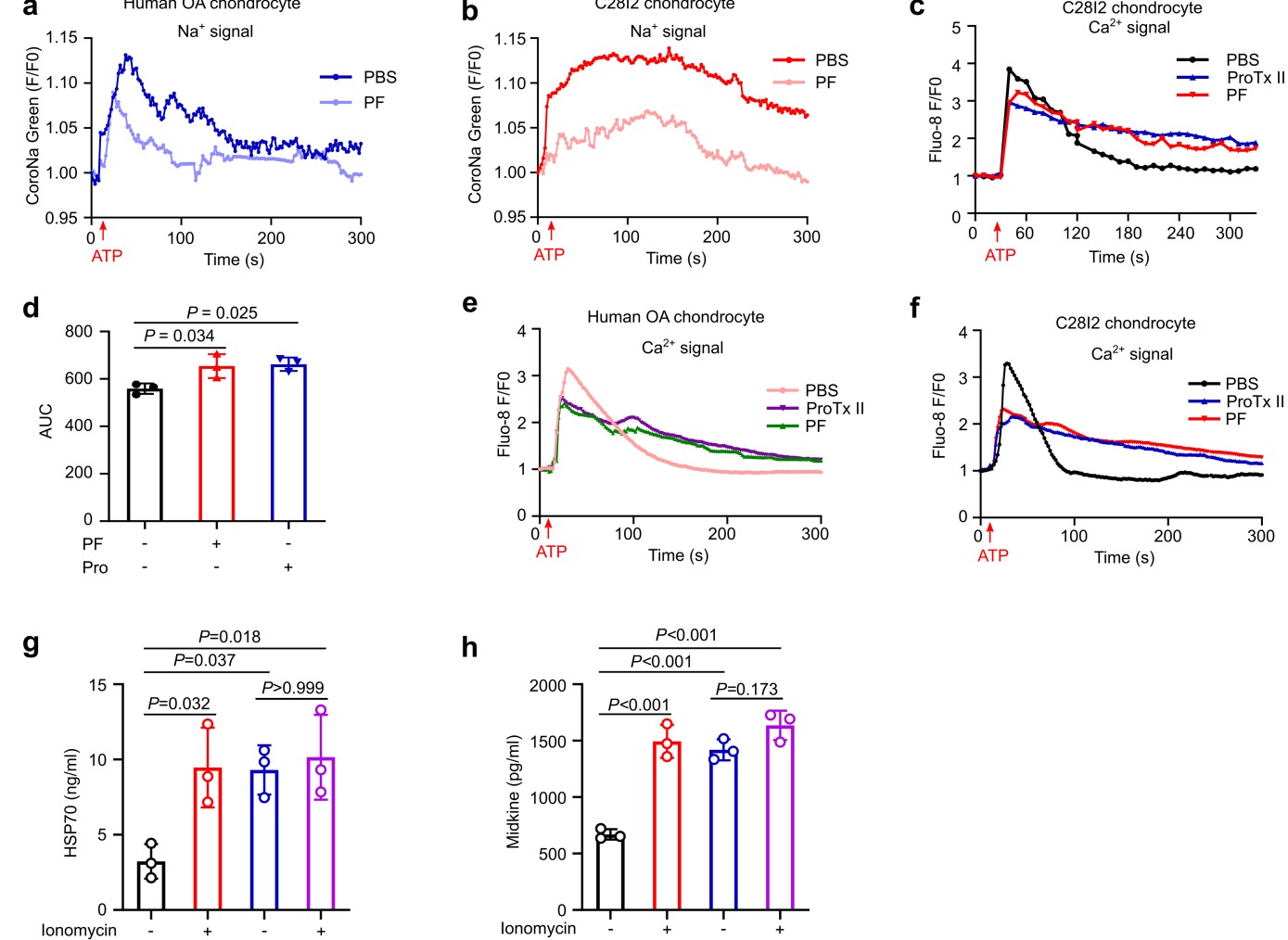

**Extended Data Fig. 12 | Intracellular Ca²⁺ signals are essential for the enhanced secretion of HSP70 and midkine following Nav1.7 blockade.** **a**, **b**, Traces (F/F0) of intracellular Na⁺ levels over time in human OA chondrocytes (**a**) and C28I2 cells (**b**) following ATP stimulation, measured by confocal microscopy. ATP present from red arrows. **c**, **d**, Traces (F/F0) (c) and quantification of AUC (d) of intracellular Ca²⁺ levels in C28I2 chondrocytes following ATP stimulation, measured by plate reader (n = 3 biological replicates). **e**, **f**, Traces (F/F0) of intracellular Ca²⁺ levels in human OA (**e**) and C28I2 (**f**) chondrocytes following ATP stimulation, assayed by confocal fluorescence microscopy. ATP present from red arrows. **g**, **h**, ELISA quantification of HSP70 (**g**) and midkine (**h**) in conditioned medium of human C28I2 cells treated with or without Ionomycin in the absence or presence of 1 μM PF-04856264 (PF) for 48 h. n = 3 biological replicates; Data are mean ± SD, *P* values are calculated by one way ANOVA with Bonferroni post-hoc test in d, g and h.

# Reporting Summary

## Statistics

For all statistical analyses, confirm that the following items are present in the figure legend, table legend, main text, or Methods section.

| n/a | Confirmed | |
|---|---|---|
| ☐ | ☒ | The exact sample size (*n*) for each experimental group/condition, given as a discrete number and unit of measurement |
| ☐ | ☒ | A statement on whether measurements were taken from distinct samples or whether the same sample was measured repeatedly |
| ☐ | ☒ | The statistical test(s) used AND whether they are one- or two-sided *Only common tests should be described solely by name; describe more complex techniques in the Methods section.* |
| ☒ | ☐ | A description of all covariates tested |
| ☐ | ☒ | A description of any assumptions or corrections, such as tests of normality and adjustment for multiple comparisons |
| ☐ | ☒ | A full description of the statistical parameters including central tendency (e.g. means) or other basic estimates (e.g. regression coefficient) AND variation (e.g. standard deviation) or associated estimates of uncertainty (e.g. confidence intervals) |
| ☐ | ☒ | For null hypothesis testing, the test statistic (e.g. *F*, *t*, *r*) with confidence intervals, effect sizes, degrees of freedom and *P* value noted *Give P values as exact values whenever suitable.* |
| ☒ | ☐ | For Bayesian analysis, information on the choice of priors and Markov chain Monte Carlo settings |
| ☒ | ☐ | For hierarchical and complex designs, identification of the appropriate level for tests and full reporting of outcomes |
| ☐ | ☒ | Estimates of effect sizes (e.g. Cohen's *d*, Pearson's *r*), indicating how they were calculated |

*Our web collection on statistics for biologists contains articles on many of the points above.*

## Software and code

Policy information about availability of computer code

| Data collection | Zeiss LSM 880 Confocal Laser Scanning Microscope (Zeiss)<br>Zeiss AXIO Microscope (Zeiss)<br>SpectraMax® i3x Multi-Mode Microplate Reader (Molecular Devices)<br>ChemiDoc Touch Imaging System (BioRad) |
|---|---|
| Data analysis | Fiji ImageJ Software (Version 2.1.0)<br>Graphpad Prism version 9<br>Image Lab 6.0.1<br>ImageJ 1.53k<br>Bio-Rad CFX manager 3.1<br>SoftMax Pro 6 (6.4.2)<br>R (version 3.5.2) |

For manuscripts utilizing custom algorithms or software that are central to the research but not yet described in published literature, software must be made available to editors and reviewers. We strongly encourage code deposition in a community repository (e.g. GitHub). See the Nature Portfolio guidelines for submitting code & software for further information.

## Data

Policy information about availability of data

  All manuscripts must include a data availability statement. This statement should provide the following information, where applicable:

- Accession codes, unique identifiers, or web links for publicly available datasets
- A description of any restrictions on data availability
- For clinical datasets or third party data, please ensure that the statement adheres to our policy

> The data supporting the findings of this study are available within the paper and supplemental information files. All material, reagents, and experimental data are available from the corresponding authors upon request. No custom code were used in this study.

## Research involving human participants, their data, or biological material

Policy information about studies with human participants or human data. See also policy information about sex, gender (identity/presentation), and sexual orientation and race, ethnicity and racism.

| | |
|---|---|
| Reporting on sex and gender | To measure HSP70 and midkine level in serum and synovial fluids, a total of 22 non-OA and 165 knee symptomatic knee OA patients from the New York biomarker cohort were included in this study according to the American college of rheumatology (ACR) criteria, the demographic information has been previously described in "Attur, M. et al. Plasma levels of interleukin-1 receptor antagonist (IL1Ra) predict radiographic progression of symptomatic knee osteoarthritis. Osteoarthritis Cartilage 23, 1915-1924, doi:10.1016/j.joca.2015.08.006 (2015)." |
| Reporting on race, ethnicity, or other socially relevant groupings | Subjects are recruited without regard to race, ethnicity, and socioecomonic status. |
| Population characteristics | Human OA cartilage samples were harvested from patients receiving total knee joint replacement surgery for OA at New York University Langone Orthopaedic Hospital. Non-arthritic femoral condyle cartilage specimens were obtained from fresh osteochondral allografts discarded following donor plug harvesting during surgical osteochondral allograft implantation. |
| Recruitment | Human cartilage were collected from de-identified donors following informed consent. |
| Ethics oversight | Human subjects research was performed according to the Institutional Review Boards at New York University Medical Center (IRB Study Number i11-01488, i9018, and i05-131). |

Note that full information on the approval of the study protocol must also be provided in the manuscript.

# Field-specific reporting

Please select the one below that is the best fit for your research. If you are not sure, read the appropriate sections before making your selection.

☒ Life sciences     ☐ Behavioural & social sciences     ☐ Ecological, evolutionary & environmental sciences

For a reference copy of the document with all sections, see [nature.com/documents/nr-reporting-summary-flat.pdf](http://nature.com/documents/nr-reporting-summary-flat.pdf)

# Life sciences study design

All studies must disclose on these points even when the disclosure is negative.

| | |
|---|---|
| Sample size | Sample sizes were not predetermined and were indicated in the figure legends. The sample size was decided based on effect sizes observed in preliminary experiments, prior experiments performed in our labs, or published findings. |
| Data exclusions | No data was excluded. |
| Replication | The number of biological replicates is as described in the figure legends. All attempts at replication were successful. |
| Randomization | For in vivo studies, age-matched mice were randomly assigned to treatment groups. For in vitro studies, cell culture experiments were pooled before splitting into individual identical wells followed by being randomly assigned to each experimental group. |
| Blinding | Experiments were performed blinded when possible. All the behavioral tests and histology evaluation were conducted in a blinded manner. For microscopy data collection and analysis, the field of view were chosen on a random basis, and were often performed by independent investigator blinded to group information, preventing biased selection of field with desired phenotype. |

# Reporting for specific materials, systems and methods

We require information from authors about some types of materials, experimental systems and methods used in many studies. Here, indicate whether each material, system or method listed is relevant to your study. If you are not sure if a list item applies to your research, read the appropriate section before selecting a response.

## Materials & experimental systems

| n/a | Involved in the study |
|---|---|
| ☐ | ☒ Antibodies |
| ☐ | ☒ Eukaryotic cell lines |
| ☒ | ☐ Palaeontology and archaeology |
| ☐ | ☒ Animals and other organisms |
| ☒ | ☐ Clinical data |
| ☒ | ☐ Dual use research of concern |
| ☒ | ☐ Plants |

## Methods

| n/a | Involved in the study |
|---|---|
| ☒ | ☐ ChIP-seq |
| ☒ | ☐ Flow cytometry |
| ☒ | ☐ MRI-based neuroimaging |

## Antibodies

| | |
|---|---|
| Antibodies used | Nav1.7 (Alomone Labs, Cat#ASC-008); TNFR2 (ProteinTech, Cat#19272-1-AP), GAPDH (ProteinTech, Cat#60004-1-Ig), NCX1 (Abcam, Cat#ab177952), Col2 (Invitrogen, Cat#MA5-12789), Mmp13 (Abcam, Cat#ab3208), aggrecan neoepitope (Novus Biologicals, Cat#NB100-74350), HSP70 (Invitrogen, Cat#MA3-009), Midkine (Abcam, Cat#ab170820) |
| Validation | All antibodies are from commercially available sources and have been validated by the manufacturers with supporting data and publications by other researchers. See below for a summary:

Nav1.7 (Alomone Labs, Cat#ASC-008)
Reactivity: Human, Mouse, Rat        Applications: ICC, IF, IHC, WB

TNFR2 (ProteinTech, Cat#19272-1-AP)
Reactivity: Human, Mouse      Applcaions: WB, IP, IF, FC, ELISA

GAPDH (ProteinTech, Cat#60004-1-Ig)
Reactivity: Human, Mouse, Rat, Yeast, Plant, Zebrafish        Applications: WB, IP, IHC, IF, FC, CoIP, ChIP, Cell treatment, ELISA

NCX1 (Abcam, Cat#ab177952)
Reactivity: Human, Mouse, Rat        Applications: WB

Col2 (Invitrogen, Cat#MA5-12789)
Reactivity: Bovine, Chicken, Human, Mouse, Rat        Applications: WB, IHC, IF, Flow

Mmp13 (Abcam, Cat#ab3208)
Reactivity: Human, Mouse, Rat        Applications: IHC, IF

aggrecan neoepitope (Novus Biologicals, Cat#NB100-74350)
Reactivity: Human, Mouse, Rat, Porcine, Bovine, Canine      Applications: IHC, IF, Flow

HSP70 (Invitrogen, Cat#MA3-009)
Reactivity: Human, Mouse        Applications: WB, IHC, Flow, Neutralization

Midkine (Abcam, Cat#ab170820)
Reactivity: Human, Mouse    Applcaions: IHC, WB
It is reported to neutralize midkine in the following publication (Xiaofan Guo, Yuan Pan, Min Xiong, Shilpa Sanapala, Corina Anastasaki, Olivia Cobb, Sonika Dahiya & David H. Gutmann. Midkine activation of CD8+ T cells establishes a neuron–immune–cancer axis responsible for low-grade glioma growth. Nature Communications volume 11, Article number: 2177, 2020.) |

## Eukaryotic cell lines

Policy information about cell lines and Sex and Gender in Research

| | |
|---|---|
| Cell line source(s) | C28I2 cell was purchased from Sigma (Cat#SCC043). Primary mouse and human chondrocytes were isolated from mouse or human cartilage. |
| Authentication | C28I2 is authenticated by Sigma. |
| Mycoplasma contamination | All cell used were mycoplasma-free. |
| Commonly misidentified lines (See ICLAC register) | No commonly misidentified cell lines were used. |

## Animals and other research organisms

Policy information about studies involving animals; ARRIVE guidelines recommended for reporting animal research, and Sex and Gender in Research

| | |
|---|---|
| Laboratory animals | C57BL/6 and Agc1-CreERT2 mice were obtained from The Jackson Laboratory (Bar Harbor, ME, USA). Nav1.8Cre;Nav1.7flox/flox mice were generously provided by Dr. John Wood at University College London, and mated with transgenic mice expressing Agc1-CreERT2 to obtain inducible Nav1.7 knockout mice in chondrocytes and both chondrocyte and DRGs. All animals were housed on a 12-hour light-dark cycle with ad libitum access to food and water in a specific pathogen-free environment. Animals were maintained on a C57BL/6J background, and age matched males typically at 12-weeks-of-age were used, if not otherwise specified in the figure legends. |
| Wild animals | This study did not involve wild animals. |
| Reporting on sex | Both female and male mice were used in the study. |
| Field-collected samples | This study did not involve samples collected from the field. |
| Ethics oversight | All animal studies were performed in accordance with institutional guidelines and approved by the Institutional Animal Care and Use Committee of New York University Grossman School of Medicine. |

Note that full information on the approval of the study protocol must also be provided in the manuscript.

