## [Peer Review File · Nature]

Manuscript Title: Nav1.7 as chondrocyte regulator and therapeutic target for osteoarthritis

Reviewer Comments & Author Rebuttals

Reviewer Reports on the Initial Version:

Referees' comments:

Referee #1 (Remarks to the Author):

This is an interesting paper describing disease modifying effects of Na channel Nav1.7 blockade in osteoarthritis.

The strengths of the manuscript include:

- Interesting observation that a channel largely associated with neuronal hypersensitivity is acting to change chondrocyte biology and has a profound effect on the disease phenotype
- Dual effect of structural modification with analgesia (acting at level of DRG)
- Strong validation in two models of OA including surgically induced OA. The size of the effects for both structural disease and pain are very large (larger than I have seen for any other intervention in these models)
- Data suggesting that commonly used drugs e.g. carbamazepine could be repurposed to test as a disease modifying drug in patients with OA

However, there are other key issues that remain unaddressed. The authors do not explain how blocking ion channels is able to influence both homeostatic and inflammatory responses in chondrocytes. Does this alter membrane signalling, bioavailability of cytokine receptors, downstream signalling? It is a strong and somewhat bizarre phenotype so understanding the mechanism is essential here. They identify two putative proteins that appear to drive the anti-catabolic and pro-anabolic effects of their drug but there is no evidence that these account for the protection that their drug or knockout offers in vivo (they would need to show that midkine or Hsp70 knockout mice were no longer protected with ion channel blockers). It is also not clear to me how the size exclusion experiments were controlled for the fact that the inhibitor would still be present in each of the fractions being assayed.

Other points:

- The analyses are all rather superficial with very simplified view of OA pathogenesis involving inflammatory cytokines and "the inflammatory environment surrounding the joint tissues". There is no direct evidence for cytokines such as IL1 driving OA despite much activity in this area over the past 2 decades.
- The authors refer to chondrocyte metabolism being affected but this is a confusing term as it might be confused for respiratory metabolism. It would be better to describe this as a change in chondrocyte biology
- Deleting Nav1.7 in chondrocytes only 4 weeks after DMM, before pain has developed and whilst there is still early disease is not testing whether the pathway in chondrocytes is analgesic. To claim this the authors would need to wait for pain to be established then show that deletion or drug inhibited pain (usually over a short time period). The only thing they can claim at present is that deletion in chondrocytes is structure-modifying.

- The pain data look very impressive but one would normally present 95% CIs rather than SDs and multiple correction should be performed rather than un-paired t tests. The authors describe the assessment as blinded but they need to indicate whether mice were randomised to drug within a cage or whether it was delivered per cage of animals. Again, this might affect introduction of bias during histological scoring.
- The authors fail to say anything about nerve growth factor, which has very strong validation as an analgesic target in mouse and human OA. Does Nav1.7 on chondrocytes influence TRKA signalling (also expressed on chondrocytes) perhaps?

Referee #2 (Remarks to the Author):

The manuscript by Fu et al studied a mouse model of osteoarthritis (OA) in search for novel disease causes and therapies. In light of pain being the major presenting symptom, the study focused on Na⁺ channels, well known as mediator of other pain syndromes including neuropathic pain. The surprising finding of the study was the expression of the Nav1.7 Na⁺ channels in both the chondrocytes that form the cartilage as well as DRG neurons that mediate the pain signal. A series of well executed experiments unequivocally identify Nav1.7 as the major culprit and provide evidence that the role of sparsely expressed Nav1.7 channels in chondrocytes regulate the secretion of proteins involved in cartilage damage. Through Mass spectrometry the authors identify two proteins, HSP70 and midkine that are specifically upregulated in the secretome of chondrocytes following Nav1.7 treatment. However, the role of these two proteins in the disease process was not further assessed by mimic or block experience which would have significantly strengthened the paper. It would have been of great interest to show that the infusion of these proteins would ameliorate cartilage damage even in the absence of Na⁺ channel blockers. Moreover, as the study uses carbamazepine (CBZ) as Na⁺ channel blocker it would have been of great interest to see if CBZ also causes enhanced release of HSP70 and midkine. Similarly, inhibiting the formation or release of the two molecules alone, for example by AAV mediated gene ablation, would be expected to induce a pain state without the induction of OA.

Hence in summary this paper presents an interesting and clinically relevant finding in an important disease that presents with chronic pain. I find the evidence for a role of Nav1.7 in chondrocytes as being mechanistically involved in OA compelling, yet the data supporting the underlying mechanism, involving changes in two proteins in the secretome incomplete.

Minor: In their electrophysiology experiments in Fig 2 the authors describe ProTx-S and TTX-S as blocking the measured Na⁺ currents. I suggest using terms such as reduce or inhibit as block insinuates a complete abolition of the current which is not the case.

Referee #3 (Remarks to the Author):

In this manuscript, Fu et al present novel findings suggesting that the voltage dependent Na channel Nav1.7 contributes to both pain and joint structural damage in osteoarthritis (OA). Genetic deletion of Nav1.7 in DRGs and/or articular chondrocytes and chemical inhibition in two mouse models of OA (DMM and MIA) supported the conclusion that targeting Nav1.7 could serve to reduce both pain and structural progression of this common and disabling condition. In vitro experiments, using a human chondrocyte cell line and, in some experiments chondrocytes from OA joints, revealed a potential mechanism by which blocking Nav1.7 resulted in increased levels of HSP70 and midkine in the chondrocyte secretome where HSP70 promoted chondrocyte anabolic gene expression and midkine inhibited catabolic gene expression. Although the findings are novel and will be of high interest to those in the OA field there are several limitations in the experimental design that require additional experiments to be done to support the conclusions of the study.

1. The destabilized medial meniscus model (DMM) and monosodium iodoacetate (MIA) mouse OA models were used for in vivo studies. The DMM model is a well-accepted OA model but the MIA model, which has been used in some past studies, does not reflect the pathobiology of human OA and has major limitations (see review of rodent OA models by Fang and Beier, Nat Rev Rheumatol, 2014). MIA is a toxin that causes cell death of joint tissue cells and a rapid onset of OA. It does not represent a means by which humans or animals develop OA. Several of the key experiments were only performed in the MIA model and not in the DMM model including the comparison of male and female mice, systemic administration of PF-04856264, and the experiments with carbamazepine. Also in the MIA model studies the intervention was given at the same time as the first injection of MIA rather than after OA had been allowed to develop as was done in the DMM model. The studies done only in the MIA model need to be repeated in the DMM model so that all experiments are done in both models. The interventions should be started at 4 weeks as was done in the other DMM experiments. The MIA results could be presented in the extended data while the main results should be from the DMM model.

2. The mouse OA experiments scored OA in cartilage and bone (including osteophytes) but the assessment of synovial changes was missing and should be added.

3. The in vitro experiments assessing Nav1.7 blockade and effects of HSP70 and midkine on anabolic and catabolic activity primarily utilized a chondrocyte cell line with only a few experiments conducted in primary cells using OA chondrocytes. These experiments were limited to analysis of gene expression in monolayer cultures. A robust assessment of chondrocyte anabolic and catabolic activity requires analysis of key factors at the protein level in primary cells in another culture system such as pellet and/or explant cultures. These additional experiments are needed to strengthen the conclusions of the study.

4. The findings that HSP-70 and midkine levels were increased in serum from OA patients (rather than being decreased) compared to controls and that both were present in OA synovial fluid does not directly support the conclusions of the study and is the opposite of what might be expected. This requires further discussion.

5. Some of the bar graphs lacked the actual datapoints which should be added.

Referee #4 (Remarks to the Author):

In this study, Fu et al. demonstrated the presence of functional Nav1.7, known to be expressed in peripheral pain-sensing neurons, in non-excitatory chondrocytes. They performed multiple assays, including RNA-seq, qPCR, Western blotting, IHC, to validate the expression of Nav1.7 in chondrocytes and its upregulation in the mouse and human OA cartilage. Further, they used electrophysiological and pharmacological blockade to disclose that the sodium current within OA chondrocytes is mainly attributed to Nav1.7, and reported that functional Nav1.7 was at channel density of 0.1-0.15 channels/ μm^2 (350-525 channels/chondrocyte). In vivo genetic loss-of-function experiments in various mouse models demonstrated that DRG neuron-expressed Nav1.7 was responsible for pain perception, whereas chondrocyte-expressed Nav1.7 regulated OA progression. Pharmacological blockade of Nav1.7 with Nav1.7-specific and clinically used pan-Nav blocker CBZ could simultaneously slow the progression of OA and reduce OA pain. Comprehensive analyses of the effects of Nav1.7 blockade on chondrocyte secretion and the corresponding conditioned medium revealed that Nav1.7 blockers regulated the chondrocyte secretome, which in turn impacted chondrocyte metabolism and OA. The combination of fractionating conditioned medium and subsequent proteomics analysis isolated HSP70 and midkine as two key molecules present in conditioned medium to regulate chondrocyte metabolism. Blocking Nav1.7 led to the increased releases of HSP70 and midkine, that stimulated anabolic and suppressed catabolic processes in chondrocytes, respectively. Overall this is a well-written paper composed of well-controlled studies. The discovery of the functional Nav1.7 in non-excitatory chondrocytes is highly innovative and impactful. These results elucidate the critical role of Nav1.7 in regulating chondrocyte metabolism and demonstrate that Nav1.7 within chondrocytes plays a previously-unrecognized important role in the progression of OA, in addition to its known controlling pain signaling in spinal sensory neurons. Considering that OA is the most common joint disease and the disease-modifying treatments for OA is currently unavailable, discovery of the important function of Nav1.7 in regulating chondrocyte metabolism and the progression of OA is exciting and of great significance from both basic (chondrocyte biology and ion channel physiology) and clinical viewpoints. The manuscript contains groundbreaking information, particularly chondrocyte expresses functional Nav1.7, that is interesting to the broad readership of arthritis, pain and sodium channels. A few points below need to be addressed.

Major comments:

1. CBZ is in widespread clinical use, has strong effects on sodium channels including Nav1.7, and is safe with acceptable side effects. Does CBZ exert a dose-dependent effects on OA progression? The authors are suggested to test a wide range of CBZ on OA model.
2. The authors examined the therapeutic effects of CBZ in chemically induced MIA model. It is suggested to further confirm its therapeutic effects on surgically induced DMM OA model as well.
3. The authors showed that PF-04856264 protected against OA progression in various WT OA models. It would be also interesting to dissect its action in OA by assessing its effects in DRG- and chondrocyte specific Nav1.7 deletion models, which allows to determine the specificity of PF-04856264 on Nav1.7.
4. Blocking Nav1.7 in chondrocytes with pharmacological inhibitors led to the increased secretion of

HSP70 and Madkine. It is suggested to test whether genetic deletion of Nav1.7 in chondrocytes also increases the release of HSP70 and Madkine into the medium.

5. Serum levels of HSP70 and Madkine are significantly elevated in OA patients compared to the healthy controls. It is suggested to examine whether this is also true in mouse DMM OA models.

Minor points:

1. The patient number of normal and OA is somehow inconsistent in Fig. 6e, g, Table S2 and figure legend. This needs to be double checked/corrected.

2. It is suggested to label the size of key bands in the DNA ladder in the Extended Fig. 1a.

Author Rebuttals to Initial Comments:

Nature manuscript 2022-11-17665

Dear Editor,

We greatly appreciate your inviting us to submit a revised version of our paper 2022-11-17665, Fu et al, titled "Nav1.7: regulator of chondrocytes and therapeutic target for osteoarthritis". We appreciate reviewers' encouraging us to strengthen this study, and have followed up with extensive new experiments as they suggested. Revisions as suggested by the reviewers, and the new data they requested, substantially enhance this paper.

Working together as a team, we have now addressed all of the reviewers' comments. The revised manuscript includes extensive additional data as requested. Point-by-point responses to the reviewer's points are given below, with the reviewer's comments presented verbatim in plain italic type, and our responses in bold:

Referee #1 (Remarks to the Author):

This is an interesting paper describing disease modifying effects of Na channel Nav1.7 blockade in osteoarthritis.

The strengths of the manuscript include:

- Interesting observation that a channel largely associated with neuronal hypersensitivity is acting to change chondrocyte biology and has a profound effect on the disease phenotype*
- Dual effect of structural modification with analgesia (acting at level of DRG)*
- Strong validation in two models of OA including surgically induced OA. The size of the effects for both structural disease and pain are very large (larger than I have seen for any other intervention in these models)*
- Data suggesting that commonly used drugs e.g. carbamazepine could be repurposed to test as a disease modifying drug in patients with OA*

However, there are other key issues that remain unaddressed. The authors do not explain how blocking ion channels is able to influence both homeostatic and inflammatory responses in chondrocytes. Does this alter membrane signalling, bioavailability of cytokine receptors, downstream signalling? It is a strong and somewhat bizarre phenotype so understanding the mechanism is essential here. They identify two putative proteins that appear to drive the anti-catabolic and pro-anabolic effects of their drug but there is no evidence that these account for the protection that their drug or knockout offers in vivo (they would need to show that midkine or Hsp70 knockout mice were no longer protected with ion channel blockers). It is also not clear to

me how the size exclusion experiments were controlled for the fact that the inhibitor would still be present in each of the fractions being assayed.

We sincerely appreciate Reviewer 1's thoughtful evaluation of our manuscript describing the disease-modifying effects of Na channel Nav1.7 blockade in OA. We are pleased that Reviewer 1 found the paper interesting and acknowledged its strengths, including the intriguing observation of a channel typically associated with neuronal hypersensitivity affecting chondrocyte biology and significantly impacting disease phenotype. We appreciate the reviewer's positive comments on the dual effect of structural modification with analgesia, strong validation in two OA models, and the potential repurposing of commonly used drugs for testing as disease-modifying agents in OA patients.

We thank Reviewer 1 for encouraging us to address the underlying mechanisms of Nav1.7 blockade's impact on chondrocytes. In response to this important comment we have carried out extensive new experiments. Specifically, we now present new data indicating that Ca²⁺ serves as a crucial second messenger, playing a pivotal role in the dynamic regulation of the chondrocyte secretome. Our new data show that (as in other non-excitabile cells such as astrocytes and microglia), Nav1.7 contributes to the regulation of intracellular Ca²⁺ signaling in chondrocytes. Building upon that finding, in the revised manuscript we present new experiments to show that Nav1.7 blockade enhances HSP70 and midkine secretion in chondrocytes through fine-tuning intracellular Ca²⁺ level which, in turn, contribute to enhanced HSP70 and midkine secretion. We show that this effect was nullified by pharmacological inhibition and genetic ablation of sodium-calcium exchanger NCX1, highlighting the crucial role of NCX1 in regulating intracellular Ca²⁺ signaling and subsequent secretion of HSP70 and midkine in response to Nav1.7 blockade in chondrocytes (Please see new Fig. 6 and Extended Data Fig. 12).

As suggested, we also carried out new experiments to more directly establish the roles of midkine and HSP70 in mediating the protective effects of Nav1.7 blockade in OA *in vivo*. Since midkine knockout mice were not commercially available, and in view of the time-frame for revision, we employed pharmaceutical inhibition using VER 155008 (HSP70 inhibitor) and iMDK (midkine inhibitor) to investigate the importance of these two proteins in mediating Nav1.7 blockade protection against OA. Specifically, we intra-articularly injected VER 155008 or iMDK, or a combination of both into DMM mice which were orally treated with the Nav1.7 blocker PF-04856264 (Please see new Fig. 5e-h). The Nav1.7 blocker PF-04856264 exhibited a notable reduction in cartilage loss and alleviation of OA-associated pain in DMM OA mice (Fig. 5e-h). However, this protective effect against OA was nearly abolished by the combined application of HSP70 inhibitor VER 155008 and midkine inhibitor iMDK (Fig. 5e-h). Of note, HSP70 inhibitor VER 155008 or midkine inhibitor iMDK alone also significantly reduced the protective effects of Nav1.7 blocker PF-04856264 against OA (Fig. 5e-h). Collectively, these findings underscore the importance of HSP70 and midkine in maintaining the protective effects of Nav1.7 blockade against OA

in vivo. We thank the reviewer for encouraging us to extend our analysis of the roles of midkine and HSP70 via these additional experiments.

We agree with the Reviewer's concern regarding the size exclusion experiments and potential interference from the inhibitor in the fractions being assayed. To address this, in the revised manuscript we now include detailed information about the procedure of this serial of exclusion experiments. Specifically, we included the molecular weights of PF-04856264 and ProTx II (437.492 and 3826.65 Da, respectively) and explained the measures taken to remove them from the relevant fraction, i.e. the fraction of <10KD. Briefly, ProTx II was depleted using Dialysis Tubing that allows the removal of molecules with molecular weights between 3.5-5KD (Micro Float-A-Lyzer, 3.5 - 5 kD, Mfr. No. F235053, Thomas Scientific), while PF-04856264 was simply removed through dialysis against the medium. We have now explicitly discussed this in the experimental section to ensure the reproducibility and reliability of our size exclusion data.

Once again, we thank Reviewer 1 for valuable feedback and constructive criticism, and for encouraging us to carry out additional experiments which expand the mechanistic reach of this work and enhance it rigor.

Other points:

- *The analyses are all rather superficial with very simplified view of OA pathogenesis involving inflammatory cytokines and "the inflammatory environment surrounding the joint tissues". There is no direct evidence for cytokines such as IL1 driving OA despite much activity in this area over the past 2 decades.*

We thank the reviewer for the point regarding the role of inflammatory cytokines in the pathogenesis of OA. We utilize IL-1 β in our *in vitro* cell models due to its extensive application in chondrocytes for mimicking *in vitro* OA conditions, and well-recognized in the field of OA research^{1,2}. Nevertheless, we acknowledge, in agreement with the reviewer, that there is a lack of direct evidence supporting IL-1 as a driver of OA. To further reveal the effects of Nav1.7 blockers on chondrocytes catabolism, we have carried out new experiment and in the revised manuscript we have now included TNF α ² and polyinosinic-polycytidilic acid (poly(I:C))³, two inflammatory inducers and also known to stimulate cartilage catabolism *in vitro* (Please see new Extended Data Fig. 7). The results demonstrated that Nav1.7 blockers also inhibited TNF α and poly(I:C) induced chondrocyte catabolism (Extended Data Fig. 7). Taken together, the findings supported the notion that Nav1.7 blockers have the potential to counteract cartilage degeneration in an inflammatory OA milieu overall.

- The authors refer to chondrocyte metabolism being affected but this is a confusing term as it might be confused for respiratory metabolism. It would be better to describe this as a change in chondrocyte biology

We thank the reviewer for urging us to be more clear. To ensure clarity and avoid misunderstanding to the readers beyond the fields of chondrocytes and OA, we have made the necessary changes, replacing it with "chondrocyte biology" wherever appropriate.

- Deleting Nav1.7 in chondrocytes only 4 weeks after DMM, before pain has developed and whilst there is still early disease is not testing whether the pathway in chondrocytes is analgesic. To claim this the authors would need to wait for pain to be established then show that deletion or drug inhibited pain (usually over a short time period). The only thing they can claim at present is that deletion in chondrocytes is structure-modifying.

We fully agree that our work shows that Nav1.7 deletion in chondrocytes is structure-modifying (i.e. that chondrocyte-expressed Nav1.7 is important for chondrocyte function, disease/structure modifying, and OA progression). We would note that, importantly, even if we perform deletion or drug inhibited pain after pain is well established, we still can not claim Nav1.7 in chondrocytes is analgesic, because it is very likely that deleting Nav1.7 in chondrocytes slows OA progression, in turn reducing OA associated pain (i.e. pain reduction can occur as a secondary effect). We fully agree with the reviewer that the only thing we should/can claim is that deletion in chondrocytes is structure-modifying, and we have modified the text to be more clear on this. We thank the reviewer for asking us to be more clear in the revision.

- The pain data look very impressive but one would normally present 95% CIs rather than SDs and multiple correction should be performed rather than un-paired t tests. The authors describe the assessment as blinded but they need to indicate whether mice were randomised to drug within a cage or whether it was delivered per cage of animals. Again, this might affect introduction of bias during histological scoring.

As suggested, the pain data was presented as means \pm 95% CI, and *P* values were calculated by multiple unpaired Student's t-test with Welch correction. In addition, all the mice were randomized to receive different drug treatment within a cage, and this pertinent information has been included in the Methods section of the revised manuscript.

- The authors fail to say anything about nerve growth factor, which has very strong validation as an analgesic target in mouse and human OA. Does Nav1.7 on chondrocytes influence TRKA signalling (also expressed on chondrocytes) perhaps?

We acknowledge the reviewer's suggestion that TRKA signaling might play a role in the effects of Nav1.7 on chondrocytes and OA. To address this possibility, we conducted experiments to assess the expressions of NGF and its receptors, TRKA and p75^{NTR}, as well as the secretion of NGF in chondrocytes treated with Nav1.7 blockers. As shown in the Figure 1 included in this letter, NGF and its receptors are both expressed in chondrocytes; however, Nav1.7 blockade did not exert any significant impact on their expressions (panels a, c, and d) or on the secretion of NGF (panel b) in chondrocytes.

Referee #2 (Remarks to the Author):

The manuscript by Fu et al studied a mouse model of osteoarthritis (OA) in search for novel diseases causes and therapies. In light of pain being the major presenting symptom, the study focused on Na⁺ channels, well known as mediator of other pain syndromes including neuropathic pain. The surprising finding of the study was the expression of the Nav1.7 Na⁺ channels in both the chondrocytes that form the cartilage as well as DRG neurons that mediate the pain signal. A series of well executed experiments unequivocally identify Nav1.7 as the major culprit and provide evidence that the role of sparsely expressed Nav1.7 channels in chondrocytes regulate the secretion of proteins involved in cartilage damage. Through Mass spectrometry the authors identify two proteins, HSP70 and midkine that are specifically upregulated in the secretome of chondrocytes following Nav1.7 treatment. However, the role of these two proteins in the disease process was not further assessed by mimic or block experience which would have significantly strengthen the paper. It would have been of great interest to show that the infusion of these proteins would ameliorate cartilage damage even in the absence of Na⁺ channel blockers. Moreover, as the study uses carbamazepine (CBZ) as Na⁺ channel blocker it would

have been of great interest to see if CBZ also causes enhanced release of HSP70 and midkine. Similarly, inhibiting the formation or release of the two molecules alone, for example by AAV mediated gene ablation, would be expected to induce a pain state without the induction of OA.

Hence in summary this paper present an interesting and clinically relevant finding in an important disease that present with chronic pain. I find the evidence for a role of Nav1.7 in chondrocytes as being mechanistically involved in OA compelling, yet the data supporting the underlying mechanism, involving changes in two proteins in the secretome incomplete.

We appreciate Reviewer 2 for considering our experiments to be well executed, and our findings to be interesting and clinically relevant in an important disease. We have performed additional experiments, as suggested, and provide new data in the revised manuscript to strengthen our conclusions. We thank the Reviewer for encouraging us to include new data that enhance this study.

We completely agree with Reviewer 2 that “*the role of these two proteins in the disease process was not further assessed by mimic or block experience which would have significantly strengthen the paper*”, and thus we carried out new blocking experiments to directly establish the roles of midkine and HSP70 in mediating the protective effects of Nav1.7 inhibitors *in vivo* (Please see new Fig. 5e-h). The Nav1.7 blocker PF-04856264 exhibited a notable reduction in cartilage loss and alleviation of OA-associated pain in DMM OA mice (Fig. 5e-h). However, this protective effect against OA was nearly abolished by the combined application of HSP70 inhibitor VER 155008 and midkine inhibitor iMDK (Fig. 5e-h). Of note, HSP70 inhibitor VER 155008 or midkine inhibitor iMDK alone also significantly reduced the protective effects of Nav1.7 blocker PF-04856264 against (Fig. 5e-h). Together, these data reveal that HSP70 and midkine are important for Nav1.7 blockade mediated protection against OA *in vivo*. Please also see the response to the main points of Reviewer 1.

We also agree that “*It would have been of great interest to show that the infusion of these proteins would ameliorate cartilage damage even in the absence of Na⁺ channel blockers*”. Indeed, overexpression of HSP70 via intra-articular injection of an adenovirus expressing HSPA1A (the gene coding HSP70) ⁴ and treatment with recombinant midkine via IP injection ⁵ have been reported to protect against OA progression. These papers have been cited in the revised manuscript. Thank you for encouraging us to be clear on this.

To determine whether CBZ also caused enhanced release of HSP70 and midkine, we performed new experiments and measured the levels of HSP70 and midkine in the medium collected from human chondrocytes treated with CBZ, and found that in consistent with the other two Nav1.7 blockers tested, CBZ also enhanced HSP70 and midkine release (this new data is included in new Extended Data Fig. 11a-d).

We also thank the reviewer for the suggestion to explore whether inhibiting these two molecules alone will affect pain state without induction of OA. We agreed that AAV mediated gene ablation would be a powerful technique that might be employed in the research. However, due to the known low transduction efficiency of AAV mediated knockdown of genes of interests in cartilage and the commercial availability of pharmacological inhibitors of HSP70 and midkine, we thus employed VER 155008 (HSP70 inhibitor) and iMDK (midkine inhibitor) to determine whether inhibition of HSP70 and midkine would affect pain state of mice without induction of pain. Specifically, Ver155008 or iMDK was intra-articularly injected into mice without OA induction, and the results demonstrated that inhibition of either of these two proteins under physiological conditions does not affect pain states (See below Fig. 2a, b), suggesting that the reduction of pain through the inhibition of HSP70 and midkine is likely to be limited to the context of Nav1.7 blockade in cases of OA, wherein both HSP70 and midkine experience a substantial increase following Nav1.7 inhibition or ablation.

Fig. 2 The effect of HSP70 or midkine inhibition on pain in mice without OA induction.

Ver155008 or iMDK was intra-articularly injected into 12-week old C57BL/6 mice every other day for a total of 4 weeks. Quantitation of 2 min travel distance (a) and von Frey testing (b) in WT mice treated with or without Ver155008 or iMDK at the indicated time-points post injection (n = 5 mice per group). *P* values are calculated by multiple unpaired T test and no statistical significance was noted for vehicle treated group vs Ver155008 or iMDK treated group. Data are means \pm SD, *P* values are calculated by one way ANOVA with Bonferroni post-hoc test.

We also greatly appreciate reviewer for “*finding the evidence for a role of Nav1.7 in chondrocytes as being mechanistically involved in OA compelling and pointing out that the data supporting the underlying mechanism, involving changes in two proteins in the secretome incomplete*”. In the revised manuscript, we have carried out extensive new experiments to address this comment. Specifically, we now present new data indicating that Ca^{2+} serves as a crucial second messenger, playing a pivotal role in the dynamic regulation of the chondrocyte secretome. In light of our new findings with both pharmacological and genetic ablation, we demonstrate that Nav1.7 plays a critical role in regulating intracellular Ca^{2+} signaling in chondrocytes through NCX1, similar to VGSCs function in other non-excitable cells like astrocytes and microglia, thereby contributing to

modulation of the secretome in chondrocytes (Please see new Fig. 6 and Extended Data Fig. 12). Please also see the response to the main points of Reviewer 1. Incorporating this data has significantly strengthened the paper. We thank the reviewer for encouraging us to extend our study in this way.

Minor: In their electrophysiology experiments in Fig 2 the authors describe ProTx-S and TTX-S as blocking the measured Na⁺ currents. I suggest using terms such as reduce or inhibit as block insinuates a complete abolition of the current which is not the case.

Agreed and changed to reduce or inhibition in the revision.

Referee #3 (Remarks to the Author):

In this manuscript, Fu et al present novel findings suggesting that the voltage dependent Na channel Nav1.7 contributes to both pain and joint structural damage in osteoarthritis (OA). Genetic deletion of Nav1.7 in DRGs and/or articular chondrocytes and chemical inhibition in two mouse models of OA (DMM and MIA) supported the conclusion that targeting Nav1.7 could serve to reduce both pain and structural progression of this common and disabling condition. In vitro experiments, using a human chondrocyte cell line and, in some experiments chondrocytes from OA joints, revealed a potential mechanism by which blocking Nav1.7 resulted in increased levels of HSP70 and midkine in the chondrocyte secretome where HSP70 promoted chondrocyte anabolic gene expression and midkine inhibited catabolic gene expression. Although the findings are novel and will be of high interest to those in the OA field there are several limitations in the experimental design that require additional experiments to be done to support the conclusions of the study.

We appreciate Reviewer 3 for considering our findings to be novel and of high interest to those in the OA field, and particularly for the suggested new assays to strengthen our conclusions.

1. The destabilized medial meniscus model (DMM) and monosodium iodoacetate (MIA) mouse OA models were used for in vivo studies. The DMM model is a well-accepted OA model but the MIA model, which has been used in some past studies, does not reflect the pathobiology of human OA and has major limitations (see review of rodent OA models by Fang and Beier, Nat Rev Rheumatol, 2014). MIA is a toxin that causes cell death of joint tissue cells and a rapid onset of OA. It does not represent a means by which humans or animals develop OA. Several of the key experiments were only performed in the MIA model and not in the DMM model including the

comparison of male and female mice, systemic administration of PF-04856264, and the experiments with carbamazepine. Also in the MIA model studies the intervention was given at the same time as the first injection of MIA rather than after OA had been allowed to develop as was done in the DMM model. The studies done only in the MIA model need to be repeated in the DMM model so that all experiments are done in both models. The interventions should be started at 4 weeks as was done in the other DMM experiments. The MIA results could be presented in the extended data while the main results should be from the DMM model.

We thank Reviewer 3 for the insightful comments. We agree and appreciate reviewer's concern about MIA model, although it is also widely used for testing the efficacy of pharmacologic agents to treat pain, as this model generates a reproducible, robust, and rapid pain-like phenotype ^{6,7}.

As suggested, we repeated all the key experiments including the comparison of male and female mice, systemic administration of PF-04856264, and the experiments with carbamazepine in surgically induced DMM model (please see new Fig. 4; Extended Data Fig. 5; Extended Data Fig. 6). The histopathological progression of the MIA model reveals loss of articular cartilage and subchondral bone lesions. As reviewer commented that MIA trigger a rapid onset of OA, cartilage lesions in the knee joint have been reported as early as day one after MIA injection ⁸. In light of this, we sincerely hope the reviewer will agree with our decision to administer the intervention concurrently with the initial MIA injection. As suggested, we have presented all the data related with MIA model in the Extended Data Figures in revision.

2. The mouse OA experiments scored OA in cartilage and bone (including osteophytes) but the assessment of synovial changes was missing and should be added.

Thank you for this suggestion. As suggested, we assessed the synovial changes in OA models and included this data in the revised manuscript (Please see new Fig. 3e, o; Extended Data Fig. 4e).

3. The in vitro experiments assessing Nav1.7 blockade and effects of HSP70 and midkine on anabolic and catabolic activity primarily utilized a chondrocyte cell line with only a few experimnts conducted in primary cells using OA chondrocytes. These experiments were limited to analysis of gene expression in monolayer cultures. A robust assessment of chondrocyte anabolic and catabolic activity requires analysis of key factors at the protein level in primary cells in another culture system such as pellet and/or explant cultures. These additional experimnts are needed to strengthen the conclusions of the study.

We thank Reviewer 3 for suggesting additional experiments with explant cultures to further strengthen the conclusion of the study. To this end, we leveraged an *ex vivo* human

OA cartilage explant assay. Full-thickness OA cartilage samples from tibia plateaus were collected from OA patients undergoing total knee arthroplasty. In line with the results obtained from in vitro monolayer culture, supernatants of the Nav1.7 blockers treated explants contained significantly higher levels of key cartilage matrix component PRG4, and lower levels of cartilage catabolic marker MMP13, compared to those of vehicle-treated controls (new Extended Data Fig. 8g, h). In addition, Nav1.7 blockers inhibited catabolism and enhanced anabolism in human cartilage explants cultured in inflammatory condition (new Extended Data Fig. 8i-j).

4. The findings that HSP-70 and midkine levels were increased in serum from OA patients (rather than being decreased) compared to controls and that both were present in OA synovial fluid does not directly support the conclusions of the study and is the opposite of what might be expected. This requires further discussion.

We thank Reviewer 3 for the comments. Our findings support the idea that Nav1.7 blockade regulates chondrocyte anabolism and catabolism through enhanced HSP70 and midkine secretion, respectively. HSP70 and midkine were also reported to exert protective effects against OA ^{4,5}. Although serum levels of HSP70 and midkine levels are elevated in OA patients compared to healthy controls, these levels might not reach therapeutic concentration necessary for substantial protection against OA progression. This paradoxical observation aligns with the evidence that some anabolic molecules like growth factors also undergo elevation alongside catabolic and inflammatory cytokines during the course of OA. As recommended, this aspect has been discussed in the Discussion section in revision. We appreciate the suggestion to be more clear on this.

5. Some of the bar graphs lacked the actual datapoints which should be added.

As suggested, we added the actual datapoints to the bar graphs.

Referee #4 (Remarks to the Author):

In this study, Fu et al. demonstrated the presence of functional Nav1.7, known to be expressed in peripheral pain-sensing neurons, in non-excitabile chondrocytes. They performed multiple assays, including RNA-seq, qPCR, Western blotting, IHC, to validate the expression of Nav1.7 in chondrocytes and its upregulation in the mouse and human OA cartilage. Further, they used electrophysiological and pharmacological blockade to disclose that the sodium current within OA chondrocytes is mainly attributed to Nav1.7, and reported that functional Nav1.7 was at channel density of 0.1-0.15 channels/ μm^2 (350-525 channels/chondrocyte). In vivo genetic loss-

of-function experiments in various mouse models demonstrated that DRG neuron-expressed Nav1.7 was responsible for pain perception, whereas chondrocyte-expressed Nav1.7 regulated OA progression. Pharmacological blockade of Nav1.7 with Nav1.7-specific and clinically used pan-Nav blocker CBZ could simultaneously slow the progression of OA and reduce OA pain. Comprehensive analyses of the effects of Nav1.7 blockade on chondrocyte secretion and the corresponding conditioned medium revealed that Nav1.7 blockers regulated the chondrocyte secretome, which in turn impacted chondrocyte metabolism and OA. The combination of fractioning conditioned medium and subsequent proteomics analysis isolated HSP70 and midkine as two key molecules present in conditioned medium to regulate chondrocyte metabolism. Blocking Nav1.7 led to the increased releases of HSP70 and midkine, that stimulated anabolic and suppressed catabolic processes in chondrocytes, respectively. Overall this is a well-written paper composed of well-controlled studies. The discovery of the functional Nav1.7 in non-excitabile chondrocytes is highly innovative and impactful. These results elucidate the critical role of Nav1.7 in regulating chondrocyte metabolism and demonstrate that Nav1.7 within chondrocytes plays a previously-unrecognized important role in the progression of OA, in addition to its known controlling pain signaling in spinal sensory neurons. Considering that OA is the most common joint disease and the disease-modifying treatments for OA is currently unavailable, discovery of the important function of Nav1.7 in regulating chondrocyte metabolism and the progression of OA is exciting and of great significance from both basic (chondrocyte biology and ion channel physiology) and clinical viewpoints. The manuscript contains groundbreaking information, particularly chondrocyte expresses functional Nav1.7, that is interesting to the broad readership of arthritis, pain and sodium channels. A few points below need to be addressed.

We are thankful to Reviewer 4 for the positive comments and for recognizing the significant value of our study, which holds importance from both basic and clinical perspectives. In the revised manuscript, we have performed additional experiments to address the reviewer's constructive questions, which have helped substantiate the conclusions.

Major comments:

1. *CBZ is in widespread clinical use, has strong effects on sodium channels including Nav1.7, and is safe with acceptable side effects. Does CBZ exert a dose-dependent effects on OA progression? The authors are suggested to test a wide range of CBZ on OA model.*

As suggested, we evaluated the potential dose-dependent effects of CBZ on OA progression in surgically induced DMM model (Extended Data Fig. 6i). DMM-operated mice treated with the low dose of CBZ (10 mg/kg body weight) exhibited substantial reduction in cartilage loss 12 weeks post-surgery, while pain sensation remained unaffected compared to untreated DMM mice (Extended Data Fig. 6j-m). The medium dose (50 mg/kg body weight) and high dose (250 mg/kg body weight) of CBZ demonstrated enhanced protection against

cartilage loss and substantial reduction in OA-associated pain than low dose CBZ (Extended Data Fig. 6j-m). We thank the reviewer for encouraging us to do these additional experiments.

2. The authors examined the therapeutic effects of CBZ in chemically induced MIA model. It is suggested to further confirm its therapeutic effects on surgically induced DMM OA model as well.

We thank the reviewer for this important suggestion. In the revised manuscript, we also examined the therapeutic effects of CBZ in surgically induced DMM model (please see new Extended Data Fig. 6i-m). Please also see the response to the question 1 of Reviewer 3.

3. The authors showed that PF-04856264 protected against OA progression in various WT OA models. It would be also interesting to dissect its action in OA by assessing its effects in DRG- and chondrocyte specific Nav1.7 deletion models, which allows to determine the specificity of PF-04856264 on Nav1.7.

To address the reviewer's comment, we have now intra-articular injected PF-04856264 into DMM operated Nav1.7^{flox} and Nav1.7^{DRG; chondrocyte} mice starting from 4 weeks after surgery for a total of 8 weeks. As shown in Figure 3 included in this letter, histological analysis revealed that Nav1.7 deletion in both DRG neurons and chondrocytes substantially attenuated cartilage loss, and reduced OARSI, and osteophyte formation and SBP thickness (Fig. 3B-E) in DMM model. The Nav1.7^{DRG; chondrocyte} mice exhibited greater overall distance of movement and significantly reduced mechanical allodynia throughout the 3 months period post DMM surgery (Fig. 3F&G). To be noted, Nav1.7 inhibitor PF-04856264 elicited indistinguishable protective effects against OA compared to vehicle treated control in Nav1.7^{DRG; chondrocyte} mice with DMM (Fig. 3B-G), indicated that PF-04856264's therapeutic effects on OA progression and OA related behavior change largely through blocking Nav1.7 in chondrocytes.

Fig. 3 (A) Schematic of experimental strategy to analyze the dependence on Nav1.7 of PF-04856264's therapeutic effects on OA progression and OA related behavior change in DMM operated Nav1.7^{fl/fl} and Nav1.7^{DRG; chondrocyte} mice. (B) Representative Safranin O/fast green stained knee joint sections of DMM operated Nav1.7^{fl/fl} and Nav1.7^{DRG; chondrocyte} mice treated with or without PF-04856264 for 8 weeks starting from 4 weeks post-surgery (n = 8 mice for each group). Scale bar = 50 μm. (C - E) Quantitation of OARSI score (C), osteophyte development (D) and SBP thickness (E) shown in B. (F, G) Quantitation of 2 min travel distance (F) and von Frey testing (G) in DMM operated Nav1.7^{fl/fl} and Nav1.7^{DRG; chondrocyte} mice treated with or without PF-04856264 at the indicated time-points after surgery (n = 8 mice per group). Data are mean ± SD, P values are calculated by one way ANOVA with Bonferroni post-hoc test.

4. Blocking Nav1.7 in chondrocytes with pharmacological inhibitors led to the increased secretion of HSP70 and Madkine. It is suggested to test whether genetic deletion of Nav1.7 in chondrocytes also increases the release of HSP70 and Madkine into the medium.

As suggested, we isolated primary chondrocytes from Nav1.7^{fl/fl} and Nav1.7^{chondrocyte} mice at 12 weeks after DMM surgery. Both HSP70 and midkine levels were up-regulated in the

conditioned medium collected from Nav1.7 KO chondrocytes compared to those from WT chondrocytes (new Fig. 5i, j).

5. Serum levels of HSP70 and Madkine are significantly elevated in OA patients compared to the healthy controls. It is suggested to examine whether this is also true in mouse DMM OA models.

As suggested, we performed new experiments to examine whether HSP70 and midkine levels are also elevated in mouse DMM OA models. To be noted, similar to what have been observed in OA patients compared to healthy controls, the serum levels of both HSP70 and midkine were markedly elevated in serum of DMM surgery mouse compared to sham surgery control (new Fig. 5k, l).

Minor points:

1. The patient number of normal and OA is somehow inconsistent in Fig. 6e, g, Table S2 and figure legend. This needs to be double checked/corrected.

We thank the reviewer for pointing into the inconsistency. We apologize for the typo in Table S2, which has been corrected in the revision.

2. It is suggested to label the size of key bands in the DNA ladder in the Extended Fig. 1a.

As suggested, we added the size of key bands in the DNA ladder in the Extended Fig. 1a.

References:

- 1 Vincent, T. L. IL-1 in osteoarthritis: time for a critical review of the literature. *F1000Res* **8** (2019). <https://doi.org:10.12688/f1000research.18831.1>
- 2 Molnar, V. *et al.* Cytokines and Chemokines Involved in Osteoarthritis Pathogenesis. *Int J Mol Sci* **22** (2021). <https://doi.org:10.3390/ijms22179208>
- 3 Li, C. *et al.* Double-stranded RNA released from damaged articular chondrocytes promotes cartilage degeneration via Toll-like receptor 3-interleukin-33 pathway. *Cell Death Dis* **8**, e3165 (2017). <https://doi.org:10.1038/cddis.2017.534>
- 4 Son, Y. O., Kim, H. E., Choi, W. S., Chun, C. H. & Chun, J. S. RNA-binding protein ZFP36L1 regulates osteoarthritis by modulating members of the heat shock protein 70 family. *Nat Commun* **10**, 77 (2019). <https://doi.org:10.1038/s41467-018-08035-7>
- 5 Xu, C. *et al.* The therapeutic effect of rhMK on osteoarthritis in mice, induced by destabilization of the medial meniscus. *Biol Pharm Bull* **37**, 1803-1810 (2014). <https://doi.org:10.1248/bpb.b14-00470>
- 6 Bove, S. E. *et al.* Weight bearing as a measure of disease progression and efficacy of anti-inflammatory compounds in a model of monosodium iodoacetate-induced osteoarthritis. *Osteoarthritis Cartilage* **11**, 821-830 (2003). [https://doi.org:10.1016/s1063-4584\(03\)00163-8](https://doi.org:10.1016/s1063-4584(03)00163-8)
- 7 Im, H. J. *et al.* Alteration of sensory neurons and spinal response to an experimental osteoarthritis pain model. *Arthritis Rheum* **62**, 2995-3005 (2010). <https://doi.org:10.1002/art.27608>
- 8 de Sousa Valente, J. The Pharmacology of Pain Associated With the Monoiodoacetate Model of Osteoarthritis. *Front Pharmacol* **10**, 974 (2019). <https://doi.org:10.3389/fphar.2019.00974>

Reviewer Reports on the First Revision:

Referees' comments:

Referee #1 (Remarks to the Author):

The authors have added further data that strengthen the mechanism behind the chondroprotection they see with blockade of Nav1.7 (through Ca signalling). They also produce evidence in vivo that this protection can be accounted for almost completely (and remarkably) by a combination of commercial inhibitors to HSP70 and Midkine.

They say that they have addressed my concerns regarding use of the word "metabolism", when meaning biology. However, this word still appears in multiple places throughout the revised manuscript and should be changed. Also, they accept that Nav1.7 blockade cannot at present be called an 'analgesia' as they haven't demonstrated that they can reverse the pain behaviour acutely (without changing structural damage). They have modified the language around this but "alleviating" and "abrogating" pain still suggests acute reversal of pain, so should be changed to "structural and symptomatic disease decrease" or an "associated reduction in pain behaviour" or the like. The authors refer to "chondrocyte secretome" regularly throughout the manuscript. Better English would be "the chondrocyte secretome". Can the authors explain why female mice have such high OA scores when so many other groups have shown that female mice are strongly protected from severe cartilage damage?

Referee #2 (Remarks to the Author):

The revised manuscript by Fu et al shows osteoarthritis employing Nav1.7 Na channels in disease pathology. The authors show that sparsely expressed Nav1.7 channels in chondrocytes regulate the secretion of proteins involved in cartilage damage which they identified through Mass spectrometry as HSP70 and midkine that are specifically upregulated in the secretome of chondrocytes following Nav1.7 treatment.

Through a series of additional experiments the authors now demonstrate that Nav1.7 activity regulates intracellular Ca²⁺ which in turn regulates release of HSP70 and midkine. They also demonstrate that these molecules are necessary and sufficient hence unequivocally establishing a mechanistic role for Nav1.7 as well as HSP70 and midkine in osteoarthritis.

The finding that inhibition of Nav1.7 can ameliorate bone loss in OA is exciting in light to the effects of a readily available; blocker, carbamazepine by the authors. Rapid translation to clinical application seems possible.

This is an important, timely and well executed study.

Harald Sontheimer

Referee #3 (Remarks to the Author):

The authors have addressed all of my concerns. The addition of the new experimental data has substantially improved the manuscript and provided the data needed to support the findings and conclusions.

Minor revision- reference #2 appears to be a reply to a comment on a review article rather than the reference to the original review.

Referee #4 (Remarks to the Author):

My questions have been addressed adequately.